# CUP: Critic-Guided Policy Reuse

**Jin Zhang[1], Siyuan Li[2], Chongjie Zhang[1]**
[1]Institute for Interdisciplinary Information Sciences, Tsinghua University, China
[2]School of Computer Science and Technology, Harbin Institute of Technology, China
jin-zhan20@mails.tsinghua.edu.cn
lisiyuan199511@gmail.com
chongjie@tsinghua.edu.cn

## Abstract

The ability to reuse previous policies is an important aspect of human intelligence. To achieve efficient policy reuse, a Deep Reinforcement Learning (DRL) agent needs to decide when to reuse and which source policies to reuse. Previous methods solve this problem by introducing extra components to the underlying algorithm, such as hierarchical high-level policies over source policies, or estimations of source policies' value functions on the target task. However, training these components induces either optimization non-stationarity or heavy sampling cost, significantly impairing the effectiveness of transfer. To tackle this problem, we propose a novel policy reuse algorithm called Critic-gUided Policy reuse (CUP), which avoids training any extra components and efficiently reuses source policies. CUP utilizes the critic, a common component in actor-critic methods, to evaluate and choose source policies. At each state, CUP chooses the source policy that has the largest one-step improvement over the current target policy, and forms a guidance policy. The guidance policy is theoretically guaranteed to be a monotonic improvement over the current target policy. Then the target policy is regularized to imitate the guidance policy to perform efficient policy search. Empirical results demonstrate that CUP achieves efficient transfer and significantly outperforms baseline algorithms.

## 1   Introduction

Human intelligence can solve new tasks quickly by reusing previous policies (Guberman & Greenfield, 1991). Despite remarkable success, current Deep Reinforcement Learning (DRL) agents lack this knowledge transfer ability (Silver et al., 2017; Vinyals et al., 2019; Ceron & Castro, 2021), leading to enormous computation and sampling cost. As a consequence, a large number of works have been studying the problem of policy reuse in DRL, i.e., how to efficiently reuse source policies to speed up target policy learning (Fernández & Veloso, 2006; Barreto et al., 2018; Li et al., 2019; Yang et al., 2020b).

A fundamental challenge towards policy reuse is: how does an agent with access to multiple source policies decide when and where to use them (Fernández & Veloso, 2006; Kurenkov et al., 2020; Cheng et al., 2020)? Previous methods solve this problem by introducing additional components to the underlying DRL algorithm, such as hierarchical high-level policies over source policies (Li et al., 2018, 2019; Yang et al., 2020b), or estimations of source policies' value functions on the target task (Barreto et al., 2017, 2018; Cheng et al., 2020). However, training these components significantly impairs the effectiveness of transfer, as hierarchical structures induce optimization non-stationarity (Pateria et al., 2021), and estimating the value functions for every source policy is computationally expensive and with high sampling cost. Thus, the objective of this study is to address the question:

*Can we achieve efficient transfer without training additional components?*

Notice that actor-critic methods (Lillicrap et al., 2016; Fujimoto et al., 2018; Haarnoja et al., 2018) learn a critic that approximates the actor's Q function and serves as a natural way to evaluate policies. Based on this observation, we propose a novel policy reuse algorithm that utilizes the critic to choose source policies. The proposed algorithm, called *Critic-gUided Policy reuse (CUP)*, avoids training any additional components and achieves efficient transfer. At each state, CUP chooses the source policy that has the largest one-step improvement over the current target policy, thus forming a guidance policy. Then CUP guides learning by regularizing the target policy to imitate the guidance policy. This approach has the following advantages. First, the one-step improvement can be estimated simply by querying the critic, and no additional components are needed to be trained. Secondly, the guidance policy is theoretically guaranteed to be a monotonic improvement over the current target policy, which ensures that CUP can reuse the source policies to improve the current target policy. Finally, CUP is conceptually simple and easy to implement, introducing very few hyper-parameters to the underlying algorithm.

We evaluate CUP on Meta-World (Yu et al., 2020), a popular reinforcement learning benchmark composed of multiple robot arm manipulation tasks. Empirical results demonstrate that CUP achieves efficient transfer and significantly outperforms baseline algorithms.

## 2   Preliminaries

Reinforcement learning (RL) deals with Markov Decision Processes (MDPs). A MDP can be modelled by a tuple $(\mathcal{S}, \mathcal{A}, r, p, \gamma)$, with state space $\mathcal{S}$, action space $\mathcal{A}$, reward function $r(s, a)$, transition function $p(s'|s, a)$, and discount factor $\gamma$ (Sutton & Barto, 2018). In this study, we focus on MDPs with continuous action spaces. RL's objective is to find a policy $\pi(a|s)$ that maximizes the cumulative discounted return $R(\pi) = \mathbb{E}_\pi \left[ \sum_{t=0}^\infty \gamma^t r(s_t, a_t) \right]$.

While CUP is generally applicable to a wide range of actor-critic algorithms, in this work we use SAC (Haarnoja et al., 2018) as the underlying algorithm. The soft Q function and soft V function (Haarnoja et al., 2017) of a policy $\pi$ are defined as:

$$Q_\pi(s, a) = r(s, a) + \gamma \mathbb{E}_{s' \sim p(\cdot|s,a)} \left[ V_\pi(s) \right] \tag{1}$$

$$V_\pi(s) = \mathbb{E}_{a \sim \pi(\cdot|s)} \left[ Q_\pi(s, a) - \alpha \log \pi(a|s) \right], \tag{2}$$

where $\alpha > 0$ is the entropy weight. SAC's loss functions are defined as:

$$
\begin{aligned}
L_{critic}(Q_\theta) &= \mathbb{E}_{(s,a,r,s') \sim \mathcal{D}} \left[ Q_\theta(s, a) - (r + \gamma V_{\overline{\theta}}(s')) \right]^2 \\
L_{actor}(\pi_\phi) &= \mathbb{E}_{s \sim \mathcal{D}} \left[ \mathbb{E}_{a \sim \pi_\phi(\cdot|s)} \left[ \alpha \log \pi_\phi(a|s) - Q_\theta(s, a) \right] \right] \\
L_{entropy}(\alpha) &= \mathbb{E}_{s \sim \mathcal{D}} \left[ \mathbb{E}_{a \sim \pi_\phi(\cdot|s)} \left[ -\alpha \log \pi_\phi(a|s) - \alpha \overline{\mathcal{H}} \right] \right],
\end{aligned}
\tag{3}
$$

where $\mathcal{D}$ is the replay buffer, $\overline{\mathcal{H}}$ is a hyper-parameter representing the target entropy, $\theta$ and $\phi$ are network parameters, $\overline{\theta}$ is target network's parameters, and $V_{\overline{\theta}}(s) = \mathbb{E}_{a \sim \pi(a|s)}[Q_{\overline{\theta}}(s, a) - \alpha \log \pi(a|s)]$ is the target soft value function.

We define the *soft expected advantage* of action probability distribution $\pi_i(\cdot|s)$ over policy $\pi_j$ at state $s$ as:

$$EA_{\pi_j}(s, \pi_i) = \mathbb{E}_{a \sim \pi_i(\cdot|s)} \left[ Q_{\pi_j}(s, a) - \alpha \log \pi_i(a|s) - V_{\pi_j}(s) \right]. \tag{4}$$

$EA_{\pi_j}(s, \pi_i)$ measures the one-step performance improvement brought by following $\pi_i$ instead of $\pi_j$ at state $s$, and following $\pi_j$ afterwards.

The field of policy reuse focuses on solving a target MDP $M$ efficiently by transferring knowledge from a set of source policies $\{\pi_1, \pi_2, ..., \pi_n\}$. We denote the target policy learned on $M$ at iteration $t$ as $\pi_{tar}^t$, and its corresponding soft Q function as $Q_{\pi_{tar}^t}$. In this work, we assume that the source policies and the target policy share the same state and action spaces.

# 3 Critic-Guided Policy Reuse

This section presents CUP, an efficient policy reuse algorithm that does not require training any additional components. CUP is built upon actor-critic methods. In each iteration, CUP uses the critic to form a guidance policy from the source policies and the current target policy. Then CUP guides policy search by regularizing the target policy to imitate the guidance policy. Section 3.1 presents how to form a guidance policy by aggregating source policies through the critic, and proves that the guidance policy is guaranteed to be a monotonic improvement over the current target policy. We also prove that the target policy is theoretically guaranteed to improve by imitating the guidance policy. Section 3.2 presents the overall framework of CUP.

## 3.1 Critic-Guided Source Policy Aggregation

CUP utilizes action probabilities proposed by source policies to improve the current target policy, and forms a guidance policy. At iteration $t$ of target policy learning, for each state $s$, the agent has access to a set of candidate action probability distributions proposed by the $n$ source policies and the current target policy: $\Pi_t^s = \{\pi_1(\cdot|s), \pi_2(\cdot|s), ..., \pi_n(\cdot|s), \pi_{tar}^t(\cdot|s)\}$. The guidance policy $\pi_g^t$ can be formed by combining the action probability distributions that have the largest soft expected advantage over $\pi_{tar}^t$ at each state $s$:

$$\pi_g^t(\cdot|s) = \underset{\pi(\cdot|s) \in \Pi_t^s}{\arg\max} EA_{\pi_{tar}^t}(s, \pi) = \underset{\pi(\cdot|s) \in \Pi_t^s}{\arg\max} \mathbb{E}_{a \sim \pi(\cdot|s)} \left[ Q_{\pi_{tar}^t}(s, a) - \alpha \log \pi(a|s) \right] \; for \; all \; s \in \mathcal{S}. \tag{5}$$

The second equation holds as adding $V_{\pi_{tar}^t}(s)$ to all soft expected advantages does not affect the result of the $\arg\max$ operator. Eq. 5 implies that at each state, we can choose which source policy to follow simply by querying its expected soft Q value under $\pi_{tar}^t$. Noticing that with function approximation, the exact soft Q value cannot be acquired. The following theorem enables us to form the guidance policy with an approximated soft Q function, and guarantees that the guidance policy is a monotonic improvement over the current target policy.

**Theorem 1** *Let $\widetilde{Q}_{\pi_{tar}^t}$ be an approximation of $Q_{\pi_{tar}^t}$ such that*

$$|\widetilde{Q}_{\pi_{tar}^t}(s, a) - Q_{\pi_{tar}^t}(s, a)| \leq \epsilon \, for \, all \, s \in \mathcal{S}, a \in A. \tag{6}$$

*Define*

$$\widetilde{\pi_g^t}(\cdot|s) = \underset{\pi(\cdot|s) \in \Pi_t^s}{\arg\max} \mathbb{E}_{a \sim \pi(\cdot|s)} \left[ \widetilde{Q}_{\pi_{tar}^t}(s, a) - \alpha \log \pi(a|s) \right] \, for \, all \, s \in \mathcal{S}. \tag{7}$$

*Then,*

$$V_{\widetilde{\pi_g^t}}(s) \geq V_{\pi_{tar}^t}(s) - \frac{2\epsilon}{1 - \gamma} \, for \, all \, s \in \mathcal{S}. \tag{8}$$

Theorem 1 provides a way to choose source policies using an approximation of the current target policy's soft Q value. As SAC learns such an approximation, the guidance policy can be formed without training any additional components.

The next question is, how to incorporate the guidance policy $\widetilde{\pi_g^t}$ into target policy learning? The following theorem demonstrates that policy improvement can be guaranteed if the target policy is optimized to stay close to the guidance policy.

**Theorem 2** *If*

$$D_{KL}\left(\pi_{tar}^{t+1}(\cdot|s) || \widetilde{\pi_g^t}(\cdot|s)\right) \leq \delta \, for \, all \, s \in \mathcal{S}, \tag{9}$$

*then*

$$V_{\pi_{tar}^{t+1}}(s) \geq V_{\pi_{tar}^t}(s) - \frac{\sqrt{2 \ln 2\delta}(\widetilde{R}_{max} + \alpha \mathcal{H}_{max}^{t+1})}{(1 - \gamma)^2} - \frac{2\epsilon + \alpha \widetilde{\mathcal{H}}_{max}}{1 - \gamma} \, for \, all \, s \in \mathcal{S}, \tag{10}$$

*where $\widetilde{R}_{max} = \underset{s,a}{\max} |r(s, a)|$ is the largest possible absolute value of the reward, $\mathcal{H}_{max}^{t+1} = \underset{s}{\max} \mathcal{H}(\pi_{tar}^{t+1}(\cdot|s))$ is the largest entropy of $\pi_{tar}^{t+1}$, and $\widetilde{\mathcal{H}}_{max} = \underset{s}{\max} \left| \mathcal{H}(\pi_{tar}^t(\cdot|s)) - \mathcal{H}(\pi_{tar}^{t+1}(\cdot|s)) \right|$ is the largest possible absolute difference of the policy entropy.*

According to Theorem 2, the target policy can be improved by minimizing the KL divergence between the target policy and the guidance policy. Thus we can use the KL divergence as an auxiliary loss to guide target policy learning. Proofs of this section are deferred to Appendix B.1 and Appendix B.2. Theorem 1 and Theorem 2 can be extended to common "hard" value functions (deferred to Appendix B.3), so CUP is also applicable to actor-critic algorithms that uses "hard" Bellman updates, such as A3C (Mnih et al., 2016).

## 3.2 CUP Framework

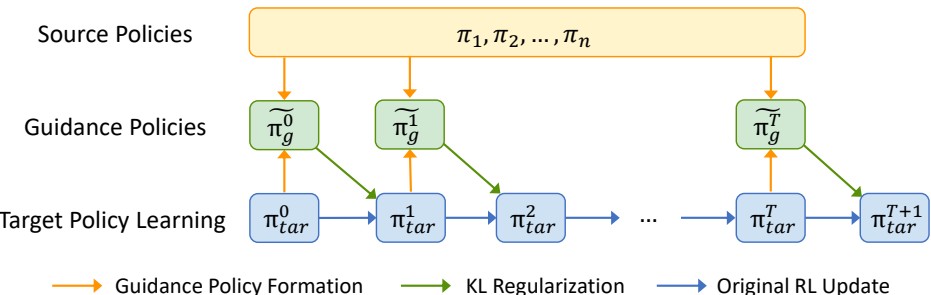

Figure 1: CUP framework. In each iteration, CUP first forms a guidance policy by querying the critic, then guides policy learning by adding a KL regularization to policy search.

In this subsection we propose the overall framework of CUP. As shown in Fig. 1, at each iteration $t$, CUP first forms a guidance policy $\widetilde{\pi_g^t}$ according to Eq. 7, then provides additional guidance to policy search by regularizing the target policy $\pi_{tar}^{t+1}$ to imitate $\widetilde{\pi_g^t}$ (Wu et al., 2019; Fujimoto & Gu, 2021). Specifically, CUP minimizes the following loss to optimize $\pi_{tar}^{t+1}$:

$$L_{CUP}(\pi_{tar}^{t+1}) = L_{actor}(\pi_{tar}^{t+1}) + \mathbb{E}_{s \sim \mathcal{D}} \left[ \beta_s D_{KL} \left( \pi_{tar}^{t+1}(\cdot|s) \| \widetilde{\pi_g^t}(\cdot|s) \right) \right], \tag{11}$$

where $L_{actor}$ is the original actor loss defined in Eq. (3), and $\beta_s > 0$ is a hyper-parameter controlling the weight of regularization. In practice, we find that using a fixed weight for regularization has two problems. First, it is difficult to balance the scale between $L_{actor}$ and the regularization term, because $L_{actor}$ grows as the Q value gets larger. Secondly, a fixed weight cannot reflect the agent's confidence on $\widetilde{\pi_g^t}$. For example, when no source policies have positive soft expected advantages, $\widetilde{\pi_g^t} = \pi_{tar}^t$. Then the agent should not imitate $\widetilde{\pi_g^t}$ anymore, as $\widetilde{\pi_g^t}$ cannot provide any guidance to further improve performance. Noticing that the soft expected advantage serves as a natural confidence measure, we weight the KL divergence with corresponding soft expected advantage at that state:

$$\beta_s = \beta_1 \min \left( \widetilde{EA}_{\pi_{tar}^t}(s, \widetilde{\pi_g^t}), \beta_2 |\widetilde{V}_{\pi_{tar}^t}(s)| \right), \tag{12}$$

where $\widetilde{EA}_{\pi_{tar}^t}(s, \widetilde{\pi_g^t}) = \mathbb{E}_{a \sim \widetilde{\pi_g^t}(\cdot|s)} \left[ \widetilde{Q}_{\pi_{tar}^t}(s, a) - \alpha \log \pi_g^t(a|s) - \widetilde{V}_{\pi_{tar}^t}(s) \right]$ is the approximated soft expected advantage, $\beta_1, \beta_2 > 0$ are two hyper-parameters, and $\widetilde{V}_{\pi_{tar}^t}(s) = \mathbb{E}_{a \sim \pi_{tar}^t(\cdot|s)} \left[ \widetilde{Q}_{\pi_{tar}^t}(s, a) - \alpha \log \pi_{tar}^t(a|s) \right]$ is the approximated soft value function. This adaptive regularization weight automatically balances between the two losses, and ignores the regularization term at states where $\widetilde{\pi_g^t}$ cannot improve over $\pi_{tar}^t$ anymore. We further upper clip the expected advantage with the absolute value of $\beta_2 \widetilde{V}_{\pi_{tar}^t}$ to avoid the agent being overly confident about $\widetilde{\pi_g^t}$ due to function approximation error $\epsilon$.

CUP's pseudo-code is presented in Alg. 1. The modifications CUP made to SAC are marked in red. Additional implementation details are deferred to Appendix D.1.

**Algorithm 1** CUP

---

**Require:** Source policies $\{\pi_1, \pi_2, ..., \pi_n\}$, hyper-parameters $\lambda_{\theta_1}, \lambda_{\theta_2}, \lambda_\pi, \lambda_\alpha, \tau, \overline{\mathcal{H}}, \textcolor{red}{\beta_1, \beta_2}$
Initialize replay buffer $\mathcal{D}$
Initialize actor $\pi_\phi$, entropy weight $\alpha$, critic $Q_{\theta_1}, Q_{\theta_2}$, target networks $Q_{\bar{\theta}_1} \leftarrow Q_{\theta_1}, Q_{\bar{\theta}_2} \leftarrow Q_{\theta_2}$
**while** not done **do**
   **for** each environment step **do**
      $a_t \sim \pi_\theta$
      $s_{t+1} \sim p(s_{t+1}|s_t, a_t)$
      $\mathcal{D} \leftarrow \mathcal{D} \cup \{s_t, a_t, r(s_t, a_t), s_{t+1}\}$
   **end for**
   **for** each gradient step **do**
      Sample minibatch $b$ from $\mathcal{D}$
      Query source policies' action probabilities $\{\pi_1(\cdot|s), \pi_2(\cdot|s), ..., \pi_n(\cdot|s)\}$ for states in $b$
      Compute expected advantages according to Eq. (4), form $\widetilde{\pi_g^t}$ according to Eq. (7)
      $\theta_i \leftarrow \theta_i - \lambda_Q \hat{\nabla}_{\theta_i} L_{critic}(Q_{\theta_i})$ for $i \in \{1, 2\}$
      $\phi \leftarrow \phi - \lambda_\pi \hat{\nabla}_\phi \textcolor{red}{L_{CUP}(\pi_\phi)}$
      $\alpha \leftarrow \alpha - \lambda_\alpha \hat{\nabla}_\alpha L_{entropy}(\alpha)$
      $\bar{\theta}_i \leftarrow \tau\theta_i + (1-\tau)\bar{\theta}_i$ for $i \in \{1, 2\}$
   **end for**
**end while**

---

# 4 Experiments

We evaluate on Meta-World (Yu et al., 2020), a popular reinforcement learning benchmark composed of multiple robot manipulation tasks. These tasks are both correlated (performed by the same Sawyer robot arm) and distinct (interacting with different objects and having different reward functions), and serve as a proper evaluation benchmark for policy reuse. The source policies are achieved by training on three representative tasks: Reach, Push, and Pick-Place. We choose several complex tasks as target tasks, including Hammer, Peg-Insert-Side, Push-Wall, Pick-Place-Wall, Push-Back, and Shelf-Place. Among these target tasks, Hammer and Peg-Insert-Side require interacting with objects unseen in the source tasks. In Push-Wall and Pick-Place-Wall, there is a wall between the object and the goal. In Push-Back, the goal distribution is different from Push. In Shelf-Place, the robot is required to put a block on a shelf, and the shelf is unseen in the source tasks. Video demonstrations of these tasks are available at `https://meta-world.github.io/`. Similar to the settings in Yang et al. (2020a), in our experiments the goal position is randomly reset at the start of every episode. Codes are available at `https://github.com/NagisaZj/CUP`.

## 4.1 Transfer Performance on Meta-World

We compare against several representative baseline algorithms, including HAAR (Li et al., 2019), PTF (Yang et al., 2020b), MULTIPOLAR (Barekatain et al., 2021), and MAMBA (Cheng et al., 2020). Among these algorithms, HAAR and PTF learn hierarchical high-level policies over source policies. MAMBA aggregates source policies' V functions to form a baseline function, and performs policy improvement over the baseline function. MULTIPOLAR learns a weighted sum of source policies' action probabilities, and learns an additional network to predict residuals. We also compare against the original SAC algorithm. All the results are averaged over six random seeds. As shown in Figure 2, CUP is the only algorithm that achieves efficient transfer on all six tasks, significantly outperforming the original SAC algorithm. HAAR has a jump-start performance on Push-Wall and Pick-Pick-Wall, but fails to further improve due to optimization non-stationarity induced by jointly training high-level and low-level policies. MULTIPOLAR achieves comparable performance on Push-Wall and Peg-Insert-Side, because the Push source policy is useful on Push-Wall (implied by HAAR's good jump-start performance), and learning residuals on Peg-Insert-Side is easier (implied by SAC's fast learning). In Pick-Place-Wall, the Pick-Place source policy is useful, but the residual is difficult to learn, so MULTIPOLAR does not work. For the remaining three tasks, the source policies are less useful, and MULTIPOLAR fails on these tasks. PTF fails as its hierarchical policy only gets updated when the agent chooses similar actions to one of the source policies, which is quite rare when

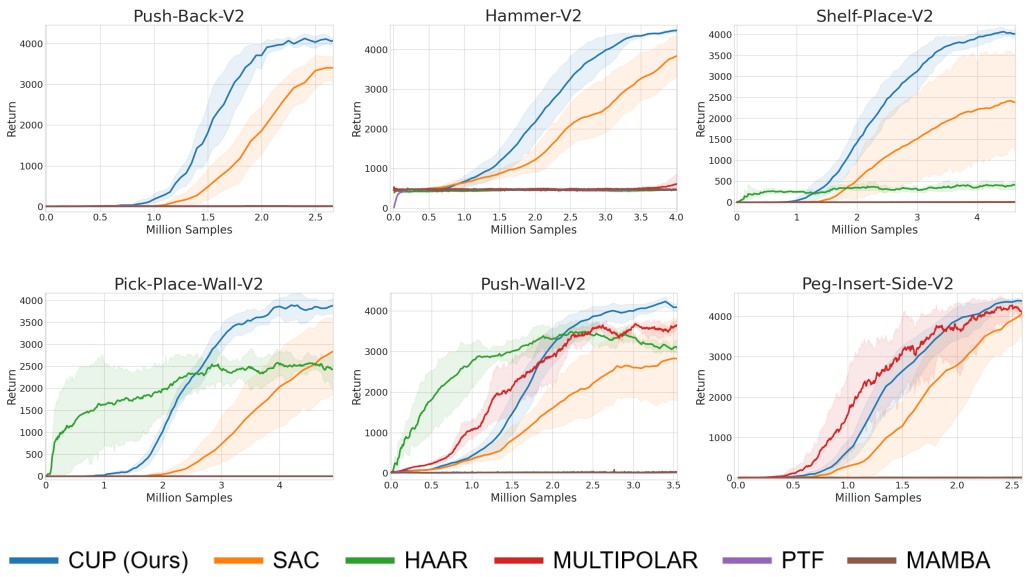

Figure 2: Evaluation of CUP and several baselines on various Meta-World tasks. Dashed areas represent 95% bootstrapped confidence intervals. CUP achieves substantially better performance than baseline algorithms.

the source and target tasks are distinct. MAMBA fails as estimating all source policies' V functions accurately is sampling inefficient. Algorithm performance evaluated by success rate is deferred to Appendix E.1.

## 4.2    Analyzing the Guidance Policy

This subsection provides visualizations of CUP's source policy selection. Fig. 3 shows the percentages of each source policy being selected throughout training on Push-Wall. At early stages of training, the source policies are selected more frequently as they have positive expected advantages, which means that they can be used to improve the current target policy. As training proceeds and the target policy becomes better, the source policies are selected less frequently. Among these three source policies, Push is chosen more frequently than the other two source policies, as it is more related to the target task. Figure 4 presents the source policies' expected advantages over an episode at convergence in Pick-Place-Wall. The Push source policy and Reach source policy almost always have negative expected advantages, which implies that these two source policies can hardly improve the current target policy anymore. Meanwhile, the Pick-Place source policy has expected advantages close to zero after 100 environment steps, which implies that the Pick-Place source policy is close to the target policy at these steps. Analyses on all six tasks as well as analyses on HAAR's source policy selection are deferred to Appendix E.2 and Appendix E.6, respectively.

## 4.3    Ablation Study

This subsection evaluates CUP's sensitivity to hyper-parameter settings and the number of source policies. We also evaluate CUP's robustness against random source policies, which do not provide meaningful candidate actions for solving target tasks.

### 4.3.1    Hyper-Parameter Sensitivity

For all the experiments in Section 4.1, we use the same set of hyper-parameters, which indicates that CUP is generally applicable to a wide range of tasks without particular fine-tuning. CUP introduces only two additional hyper-parameters to the underlying SAC algorithm, and we further test CUP's sensitivity to these additional hyper-parameters. As shown in Fig. 5, CUP is generally robust to the choice of hyper-parameters and achieves stable performance.

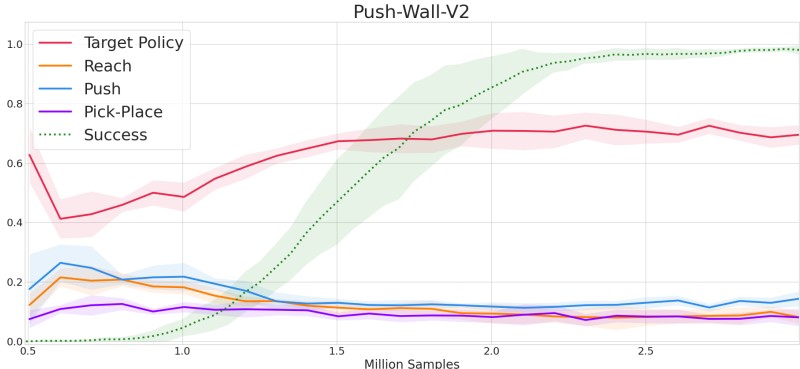

Figure 3: Percentages of source policies being selected by CUP during training on Push-Wall. The green dashed line represents the target policy's success rate on the task.

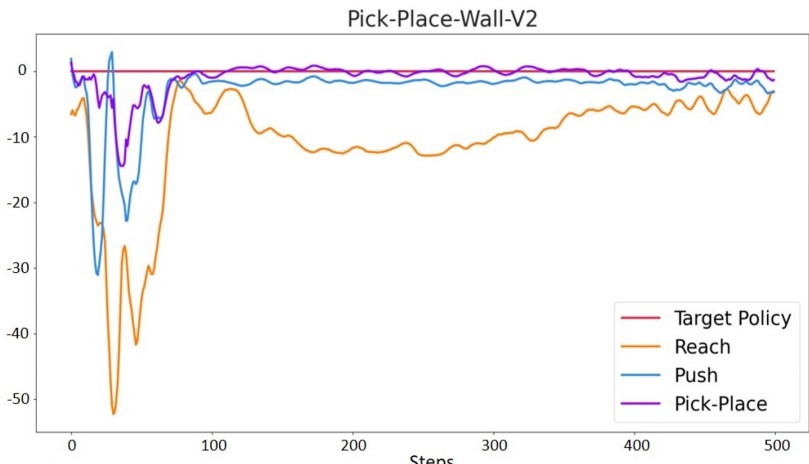

Figure 4: Expected advantages of source policies at convergence on Pick-Place-Wall. The horizontal axis represents the environment steps of an episode.

### 4.3.2 Number of Source Policies

We evaluate CUP as well as baseline algorithms on a larger source policy set. We add three policies to the original source policy set, which solve three simple tasks including Drawer-Close, Push-Wall, and Coffee-Button. This forms a source policy set composed of six policies. As shown in Fig. 6, CUP is still the only algorithm that solves all the six target tasks efficiently. MULTIPOLAR suffers from a decrease in performance, which indicates that learning the weighted sum of source policies' actions becomes more difficult as the number of source policies grows. The rest of the baseline algorithms have similar performance to those using three source policies. Fig. 7 provides a more direct comparison of CUP's performance with different number of source policies. CUP is able to utilize the additional source policies to further improve its performance, especially on Pick-Place-Wall and Peg-Insert-Side. Further detailed analysis is deferred to Appendix E.3.

### 4.3.3 Interference of Random Source Policies

In order to evaluate the efficiency of CUP's critic-guided source policy aggregation, we add random policies to the set of source policies. As shown in Fig. 8(a), adding up to 3 random source policies does not affect CUP's performance. This indicates that CUP can efficiently choose which source policy to follow even if there exist many source policies that are not meaningful. Adding 4 and 5 random source policies leads to a slight drop in performance. This drop is because that as the number

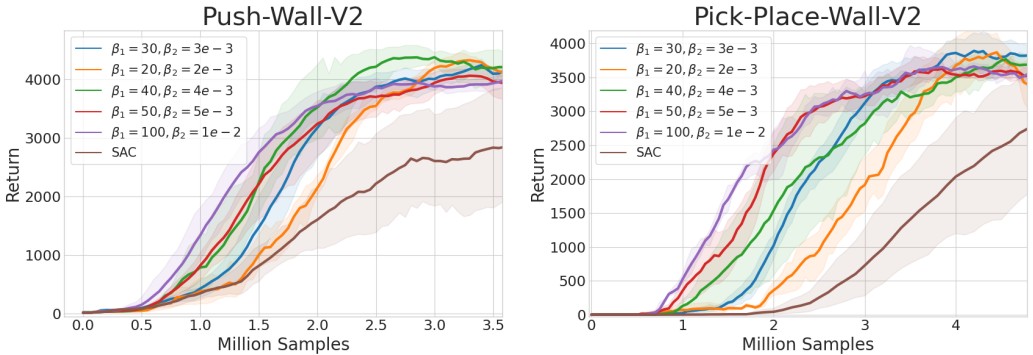

Figure 5: Ablation studies on a wide range of hyper-parameters. CUP performs well on a wide range of hyper-parameters.

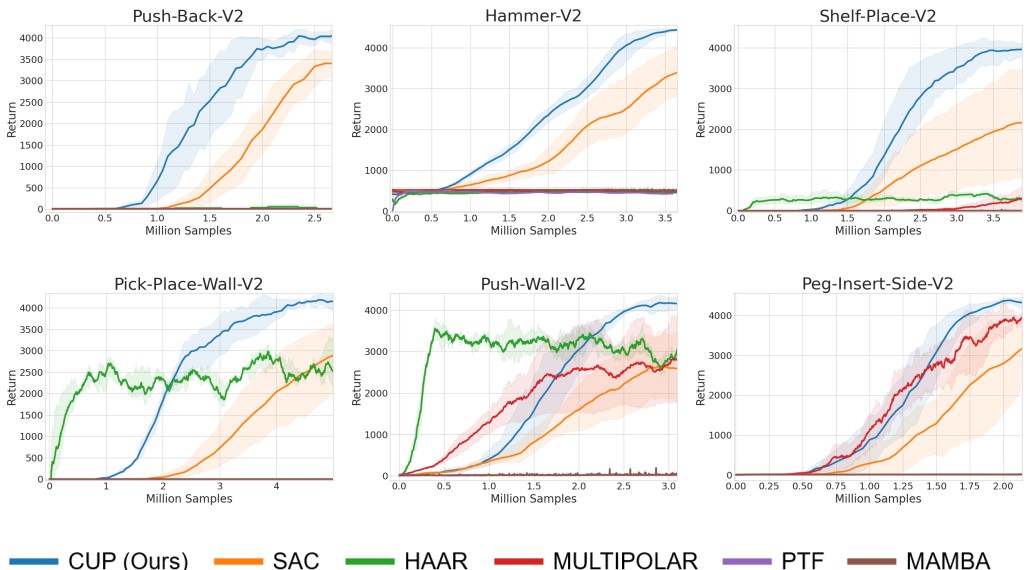

Figure 6: Performance of CUP and baseline algorithms on various Meta-World tasks, with a set of six source policies.

of random policies grows, more random actions are sampled, and taking argmax over these actions' expected advantages is more likely to be affected by errors in value estimation.

To further investigate CUP's ability to ignore unsuitable source policies, we design another transfer setting that consists of another two source policy sets. The first set consists of three random policies that are useless for the target task, and the second set adds the Reach policy to the first set. As demonstrated in Fig. 8(b), when none of the source policies are useful, CUP performs similarly to the original SAC, and its sample efficiency is almost unaffected by the useless source policies. When there exists a useful source policy, CUP can efficiently utilize it to improve performance, even if there are many useless source policies.

## 5 Related Work

**Policy reuse.** A series of works on policy reuse utilize source policies for exploration in value-based algorithms (Fernández & Veloso, 2006; Li & Zhang, 2018; Gimelfarb et al., 2021), but they are not applicable to policy gradient methods due to the off-policyness problem (Fujimoto et al., 2019). AC-Teach (Kurenkov et al., 2020) mitigates this problem by improving the actor over behavior policy's

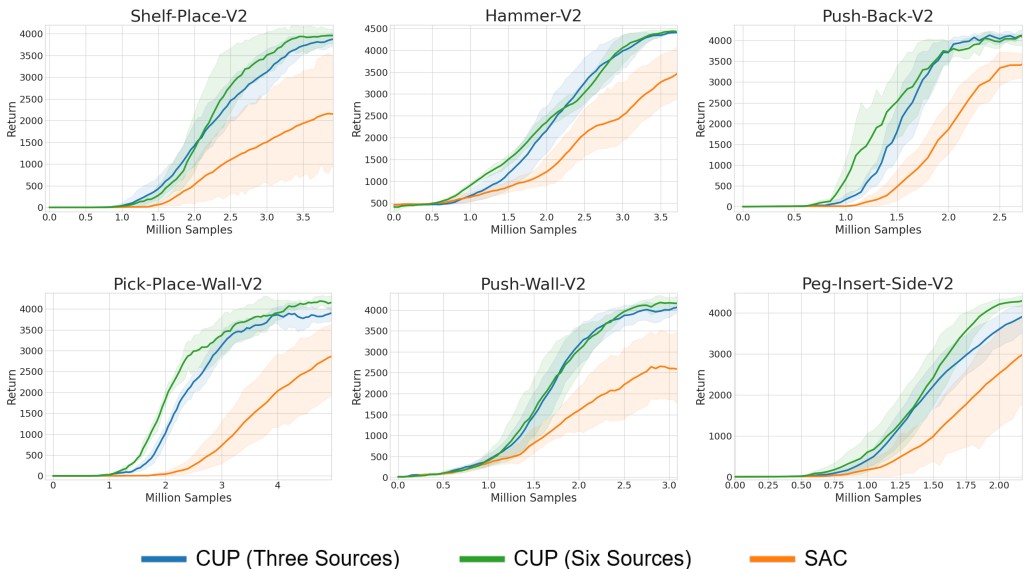

Figure 7: Comparison of CUP's performance with different number of source policies.

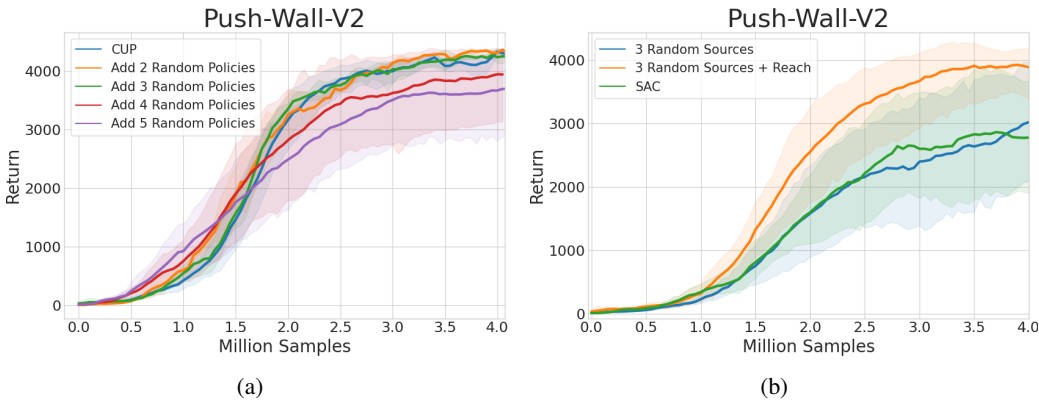

Figure 8: Ablation studies on CUP's sensitivity to useless source policies. (a) Adding up to 3 random policies to the source policy set does not affect CUP's performance. (b) Ablation study in a setting where most source policies are useless. If none of the source policies are useful (3 Random Sources), CUP performs similarly to the original SAC. Even if only one of the four source policies is useful (3 Random Sources+Reach), CUP is still able to efficiently utilize the useful source policy to improve learning performance.

value estimation, but still fails in more complex tasks. One branch of methods train hierarchical high-level policies over source policies. CAPS (Li et al., 2018) guarantees the optimality of the hierarchical policies by adding primitive skills to the low-level policy set, but is inapplicable to MDPs with continuous action spaces. HAAR (Li et al., 2019) fine-tunes low-level policies to ensure optimality, but joint training of high-level and low-level policies induce optimization non-stationarity (Pateria et al., 2021). PTF (Yang et al., 2020b) trains a hierarchical policy, which is imitated by the target policy. However, the hierarchical policy only gets updated when the target policy chooses similar actions to one of the source policies, so PTF fails in complex tasks with large action spaces. Another branch of works aggregate source policies via their Q functions or V functions on the target task. Barreto et al. (2017) and Barreto et al. (2018) focus on the situation where source tasks and target tasks share the same dynamics, and aggregate source policies by choosing the policy that has the largest Q at each state. They use successor features to mitigate the heavy computation cost

brought by estimating Q functions for all source policies. MAMBA (Cheng et al., 2020) forms a baseline function by aggregating source policies' V functions, and guides policy search by improving the policy over the baseline function. Finally, MULTIPOLAR (Barekatain et al., 2021) learns a weighted sum over source policies' actions, and learns an auxiliary network to predict residuals around the aggregated actions. MULTIPOLAR is computationally expensive, as it requires querying all the source policies at every sampling step. Our proposed method, CUP, focuses on the setting of learning continuous-action MDPs with actor-critic methods. CUP is both computationally and sampling efficient, as it does not require training any additional components.

**Policy regularization.** Adding regularization to policy optimization is a common approach to induce prior knowledge into policy learning. Distral (Teh et al., 2017) achieves inter-task transfer by imitating an average policy distilled from policies of related tasks. In offline RL, policy regularization serves as a common technique to keep the policy close to the behavior policy used to collect the dataset (Wu et al., 2019; Nair et al., 2020; Fujimoto & Gu, 2021). CUP uses policy regularization as a means to provide additional guidance to policy search with the guidance policy.

## 6 Conclusion

In this study, we address the problem of reusing source policies without training any additional components. By utilizing the critic as a natural evaluation of source policies, we propose CUP, an efficient policy reuse algorithm without training any additional components. CUP is conceptually simple, easy to implement, and has theoretical guarantees. Empirical results demonstrate that CUP achieves efficient transfer on a wide range of tasks. As for future work, CUP assumes that all source policies and the target policy share the same state and action spaces, which limits CUP's application to more general scenarios. One possible future direction is to take inspiration from previous works that map the state and action spaces of an MDP to another MDP with similar high-level structure (Wan et al., 2020; Zhang et al., 2020; Heng et al., 2022; van der Pol et al., 2020b,a). Another interesting direction is to incorporate CUP into the continual learning setting (Rolnick et al., 2019; Khetarpal et al., 2020), in which an agent gradually enriches its source policy set in an online manner.

## Acknowledgements

This work is supported in part by Science and Technology Innovation 2030 – "New Generation Artificial Intelligence" Major Project (No. 2018AAA0100904), National Natural Science Foundation of China (62176135), and China Academy of Launch Vehicle Technology (CALT2022-18).

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
