## A    Broader Social Impact

We believe policy reuse serves as a promising way to transfer knowledge among AI agents. This ability will enable AI agents to master new skills efficiently. However, we are also aware of possible negative social impacts, such as plagiarizing other AI products by querying and reusing their policies.

## B    Proofs

### B.1    Proof for Theorem 1

*Proof.* As $|\widetilde{Q}_{\pi_{tar}^t}(s,a) - Q_{\pi_{tar}^t}(s,a)| \leq \epsilon$ for all $s \in \mathcal{S}, a \in A$, we have that for all $s_i \in \mathcal{S}$, the difference between the true value function $V_{\pi_{tar}^t}$ and the approximated value function $\widetilde{V}_{\pi_{tar}^t}$ is

bounded:

$$
\begin{aligned}
&V_{\pi_{tar}^t}(s_i) \\
&= \mathbb{E}_{a_i \sim \pi_{tar}^t(\cdot|s_i)} \left[ Q_{\pi_{tar}^t}(s_i, a_i) - \alpha \log \pi_{tar}^t(a_i|s_i) \right] \\
&\leq \mathbb{E}_{a_i \sim \pi_{tar}^t(\cdot|s_i)} \left[ \widetilde{Q}_{\pi_{tar}^t}(s_i, a_i) - \alpha \log \pi_{tar}^t(a_i|s_i) + \epsilon \right] \\
&= \widetilde{V}_{\pi_{tar}^t}(s_i) + \epsilon.
\end{aligned}
$$

As $\pi_{tar}^t(\cdot|s)$ is contained in $\Pi_t^s$, with $\widetilde{\pi_g^t}$ defined in Eq. (7), it is obvious that for all $s \in \mathcal{S}$, $\mathbb{E}_{a \sim \widetilde{\pi_g^t}(\cdot|s)} \left[ \widetilde{Q}_{\pi_{tar}^t}(s, a) - \alpha \log \widetilde{\pi_g^t}(a|s) \right] \geq \mathbb{E}_{a \sim \pi_{tar}^t(\cdot|s)} \left[ \widetilde{Q}_{\pi_{tar}^t}(s, a) - \alpha \log \pi_{tar}^t(a|s) \right] = \widetilde{V}_{\pi_{tar}^t}(s)$. Then for all $s_i \in \mathcal{S}$,

$$
\begin{aligned}
&V_{\pi_{tar}^t}(s_i) \\
&\leq \widetilde{V}_{\pi_{tar}^t}(s_i) + \epsilon \\
&\leq \mathbb{E}_{a_i \sim \widetilde{\pi_g^t}(\cdot|s_i)}[\widetilde{Q}_{\pi_{tar}^t}(s_i, a_i) - \alpha \log \widetilde{\pi_g^t}(a_i|s_i)] + \epsilon \\
&\leq \mathbb{E}_{a_i \sim \widetilde{\pi_g^t}(\cdot|s_i)}[Q_{\pi_{tar}^t}(s_i, a_i) - \alpha \log \widetilde{\pi_g^t}(a_i|s_i)] + 2\epsilon \\
&= \mathbb{E}_{a_i \sim \pi_g^t(a|s_i)}[r(s_i, a_i) - \alpha \log \widetilde{\pi_g^t}(a_i|s_i) + \gamma V_{\pi_{tar}^t}(s_{i+1})] + 2\epsilon \\
&\vdots \\
&\leq \mathbb{E}_{\widetilde{\pi_g^t}}[\sum_{\tau=0}^{\infty} \gamma^\tau (r(s_{i+\tau}, a_{i+\tau}) - \alpha \log \widetilde{\pi_g^t}(a_{i+\tau}|s_{i+\tau}))] + 2 \sum_{\tau=0}^{\infty} \gamma^\tau \epsilon \\
&= V_{\widetilde{\pi_g^t}}(s_i) + \frac{2\epsilon}{1 - \gamma}.
\end{aligned}
$$

## B.2 Proof of Theorem 2

*Proof.* According to Pinsker's inequality (Fedotov et al., 2003), $D_{KL}(\pi_{tar}^{t+1}(\cdot|s)||\widetilde{\pi_g^t}(\cdot|s)) \geq \frac{1}{2\ln 2}||\pi_{tar}^{t+1}(\cdot|s) - \widetilde{\pi_g^t}(\cdot|s)||_1^2$, where $|| \cdot ||_1$ is the L1 norm. So we have that for all $s \in \mathcal{S}$, $||\pi_{tar}^{t+1}(\cdot|s) - \widetilde{\pi_g^t}(\cdot|s)||_1 \leq \sqrt{2\ln 2\delta}$. According to the Performance Difference Lemma (Kakade &

Langford, 2002), we have that for all $s \in \mathcal{S}$:

$$V_{\widetilde{\pi}_g^t}(s) - V_{\pi_{tar}^{t+1}}(s)$$

$$= \frac{1}{1-\gamma} \mathbb{E}_{s' \sim \mu_s^{\widetilde{\pi}_g^t}(s')}$$

$$\left[ \mathbb{E}_{a \sim \widetilde{\pi}_g^t(\cdot|s')} [Q_{\pi_{tar}^{t+1}}(s',a) - \alpha \log \widetilde{\pi}_g^t(a|s)] - \mathbb{E}_{a \sim \widetilde{\pi_{tar}^{t+1}}(\cdot|s')} [Q_{\pi_{tar}^{t+1}}(s',a) - \alpha \log \widetilde{\pi_{tar}^{t+1}}(a|s)] \right]$$

$$\leq \frac{1}{1-\gamma} \max_{s' \in \mathcal{S}} \left[ \mathbb{E}_{a \sim \widetilde{\pi}_g^t(\cdot|s')} [Q_{\pi_{tar}^{t+1}}(s',a)] - \mathbb{E}_{a \sim \pi_{tar}^{t+1}(\cdot|s')} [Q_{\pi_{tar}^{t+1}}(s',a)] \right]$$

$$+ \frac{\alpha}{1-\gamma} \max_{s'' \in \mathcal{S}} \left| \mathcal{H}(\widetilde{\pi}_g^t(\cdot|s'')) - \mathcal{H}(\pi_{tar}^t(\cdot|s'')) \right|$$

$$= \frac{1}{1-\gamma} \max_{s' \in \mathcal{S}} \int \left( \widetilde{\pi}_g^t(\cdot|s) - \pi_{tar}^{t+1}(a|s) \right) Q_{\pi_{tar}^{t+1}}(s',a) da + \frac{\alpha}{1-\gamma} \widetilde{\mathcal{H}}_{max}$$

$$\leq \frac{1}{1-\gamma} \max_{s' \in \mathcal{S}} \int \left| \widetilde{\pi}_g^t(a|s) - \pi_{tar}^{t+1}(a|s) \right| \cdot \left| Q_{\pi_{tar}^{t+1}}(s',a) \right| da + \frac{\alpha}{1-\gamma} \widetilde{\mathcal{H}}_{max}$$

$$\leq \frac{1}{1-\gamma} \max_{s' \in \mathcal{S}} \int \left| \widetilde{\pi}_g^t(a|s) - \pi_{tar}^{t+1}(a|s) \right| \cdot \frac{\widetilde{R}_{max} + \alpha \mathcal{H}_{max}^{t+1}}{1-\gamma} da + \frac{\alpha}{1-\gamma} \widetilde{\mathcal{H}}_{max}$$

$$= \frac{\widetilde{R}_{max} + \alpha \mathcal{H}_{max}^{t+1}}{(1-\gamma)^2} \max_{s' \in \mathcal{S}} ||\widetilde{\pi}_g^t(\cdot|s) - \pi_{tar}^{t+1}(\cdot|s)||_1 + \frac{\alpha}{1-\gamma} \widetilde{\mathcal{H}}_{max}$$

$$\leq \frac{\sqrt{2 \ln 2 \delta}(\widetilde{R}_{max} + \alpha \mathcal{H}_{max}^{t+1}) + \alpha(1-\gamma)\widetilde{\mathcal{H}}_{max}}{(1-\gamma)^2},$$

$$\tag{13}$$

where $\mu_s^{\widetilde{\pi}_g^t}(s') = (1-\gamma) \sum_{t=0}^{\infty} \gamma^t p(s_t = s'|s_0 = s, \widetilde{\pi}_g^t)^1$ is the normalized discounted state occupancy distribution. Note that

$$|Q_{\pi_{tar}^{t+1}}(s,a)|$$

$$= \left| \mathbb{E}_{\pi_{tar}^{t+1}} \left[ \sum_{i=0}^{\infty} \gamma^i (r(s_{\tau+i}, a_{\tau+i}) - \alpha \log \pi_{tar}^{t+1}(\cdot|s))|s_\tau = s, a_\tau = a \right] \right|$$

$$\leq \mathbb{E}_{\pi} \left[ \sum_{i=0}^{\infty} \gamma^i (\widetilde{R}_{max} + \gamma \mathcal{H}_{max}^{t+1}) \right] \tag{14}$$

$$= \frac{\widetilde{R}_{max} + \alpha \mathcal{H}_{max}^{t+1}}{1-\gamma}. \tag{15}$$

Eventually, we have $V_{\pi_{tar}^{t+1}}(s) \geq V_{\widetilde{\pi}_g^t}(s) - \frac{\sqrt{2 \ln 2 \delta}(\widetilde{R}_{max}+\alpha \mathcal{H}_{max}^{t+1})+\alpha(1-\gamma)\widetilde{\mathcal{H}}_{max}}{(1-\gamma)^2} \geq V_{\pi_{tar}^t}(s) - \frac{\sqrt{2 \ln 2 \delta}(\widetilde{R}_{max}+\alpha \mathcal{H}_{max}^{t+1})}{(1-\gamma)^2} - \frac{2\epsilon + \alpha \widetilde{\mathcal{H}}_{max}}{1-\gamma}$.

## B.3 Critic-Guided Source Policy Aggregation under "Hard" Value Functions

In this section we override the notation $Q$, $V$ to represent "hard" value functions, and override the notation $EA$ to represent the *expected advantage*, which is defined as $EA_{\pi_j}(s, \pi_i) = \mathbb{E}_{a \sim \pi_i(\cdot|s)} \left[ Q_{\pi_j}(s,a) - V_{\pi_j}(s) \right]$. Then Theorem 1 and Theorem 2 can be extended as below.

**Theorem 3** *Let $\widetilde{Q}_{\pi_{tar}^t}$ be an approximation of $Q_{\pi_{tar}^t}$ such that*

$$|\widetilde{Q}_{\pi_{tar}^t}(s,a) - Q_{\pi_{tar}^t}(s,a)| \leq \epsilon \text{ for all } s \in \mathcal{S}, a \in A. \tag{16}$$

*Define*

$$\widetilde{\pi}_g^t(\cdot|s) = \arg\max_{\pi(\cdot|s) \in \Pi_t^s} \mathbb{E}_{a \sim \pi(\cdot|s)} \left[ \widetilde{Q}_{\pi_{tar}^t}(s,a) \right] \text{ for each } s \in \mathcal{S}. \tag{17}$$

---

[1] We slightly abuse the notation $s_0$ here to indicate that the agent start deterministically from state $s$.

*Then,*

$$V_{\widetilde{\pi}_g^t}(s) \geq V_{\pi_{tar}^t}(s) - \frac{2\epsilon}{1-\gamma} \text{ for all } s \in \mathcal{S}. \tag{18}$$

**Theorem 4** *If*

$$D_{KL}\left(\pi_{tar}^{t+1}(\cdot|s)||\widetilde{\pi}_g^t(\cdot|s)\right) \leq \delta \text{ for all } s \in \mathcal{S}, \tag{19}$$

*then*

$$V_{\pi_{tar}^{t+1}}(s) \geq V_{\pi_{tar}^t}(s) - \frac{\sqrt{2\ln 2\delta}\widetilde{R}_{max}}{(1-\gamma)^2} - \frac{2\epsilon}{1-\gamma} \text{ for all } s \in \mathcal{S}, \tag{20}$$

*where $\widetilde{R}_{max} = \max\limits_{s,a} |r(s,a)|$ is the largest possible absolute value of the reward.*

Theorem 3 and Theorem 4 implies that CUP can still guarantee policy improvement under hard Bellman updates. Proofs are given below.

**Proof for Theorem 3.** As $|\widetilde{Q}_{\pi_{tar}^t}(s,a) - Q_{\pi_{tar}^t}(s,a)| \leq \epsilon$ for all $s \in \mathcal{S}, a \in A$, we have that for all $s_i \in \mathcal{S}$, the difference between the true value function $V_{\pi_{tar}^t}$ and the approximated value function $\widetilde{V}_{\pi_{tar}^t}$ is bounded:

$$
\begin{aligned}
&V_{\pi_{tar}^t}(s_i) \\
&= \mathbb{E}_{a \sim \pi_{tar}^t(\cdot|s)}\left[Q_{\pi_{tar}^t}(s_i, a_i)\right] \\
&\leq \mathbb{E}_{a \sim \pi_{tar}^t(\cdot|s)}\left[\widetilde{Q}_{\pi_{tar}^t}(s_i, a_i) + \epsilon\right] \\
&= \widetilde{V}_{\pi_{tar}^t}(s_i) + \epsilon.
\end{aligned}
$$

As $\pi_{tar}^t(\cdot|s)$ is contained in $\Pi_t^s$, with $\pi_g^t$ defined in Eq. (17), it is obvious that for all $s \in \mathcal{S}$, $\mathbb{E}_{a \sim \widetilde{\pi}_g^t(\cdot|s)}\left[\widetilde{Q}_{\pi_{tar}^t}(s,a)\right] \geq \mathbb{E}_{a \sim \pi_{tar}^t(\cdot|s)}\left[\widetilde{Q}_{\pi_{tar}^t}(s,a)\right] = \widetilde{V}_{\pi_{tar}^t}(s)$. Then for all $s_i \in \mathcal{S}$,

$$
\begin{aligned}
&V_{\pi_{tar}^t}(s_i) \\
&\leq \widetilde{V}_{\pi_{tar}^t}(s_i) + \epsilon \\
&\leq \mathbb{E}_{a_i \sim \widetilde{\pi}_g^t(\cdot|s_i)}[\widetilde{Q}_{\pi_{tar}^t}(s_i, a_i)] + \epsilon \\
&\leq \mathbb{E}_{a_i \sim \widetilde{\pi}_g^t(\cdot|s_i)}[Q_{\pi_{tar}^t}(s_i, a_i)] + 2\epsilon \\
&= \mathbb{E}_{a_i \sim \pi_g^t(a|s_i)}[r(s_i, a_i) + \gamma V_{\pi_{tar}^t}(s_{i+1})] + 2\epsilon \\
&\leq \mathbb{E}_{a_i \sim \widetilde{\pi}_g^t(\cdot|s_i)}[r(s_i, a_i) + \gamma(\widetilde{V}_{\pi_{tar}^t}(s_{i+1}) + \epsilon)] + 2\epsilon \\
&\leq \mathbb{E}_{a_i \sim \widetilde{\pi}_g^t(\cdot|s_i), a_{i+1} \sim \widetilde{\pi}_g^t(\cdot|s_{i+1})}[r(s_i, a_i) + \gamma\widetilde{Q}_{\pi_{tar}^t}(s_{i+1}, a_{i+1})] + (2+\gamma)\epsilon \\
&\leq \mathbb{E}_{a_i \sim \widetilde{\pi}_g^t(\cdot|s_i), a_{i+1} \sim \widetilde{\pi}_g^t(\cdot|s_{i+1})}[r(s_i, a_i) + \gamma(Q_{\pi_{tar}^t}(s_{i+1}, a_{i+1}) + \epsilon)] + (2+\gamma)\epsilon \\
&= \mathbb{E}_{a_i \sim \widetilde{\pi}_g^t(\cdot|s_i), a_{i+1} \sim \widetilde{\pi}_g^t(\cdot|s_{i+1})}[r(s_i, a_i) + \gamma r(s_{i+1}, a_{i+1}) + \gamma^2 V_{\pi_{tar}^t}(s_{i+2})] + (2+2\gamma)\epsilon \\
&\vdots \\
&\leq \mathbb{E}_{\widetilde{\pi}_g^t}[\sum_{\tau=0}^{\infty} \gamma^\tau r(s_{i+\tau}, a_{i+\tau})] + 2\sum_{\tau=0}^{\infty} \gamma^\tau \epsilon \\
&= V_{\widetilde{\pi}_g^t}(s_i) + \frac{2\epsilon}{1-\gamma}.
\end{aligned}
$$

**Proof for Theorem 4.** According to Pinsker's inequality (Fedotov et al., 2003), $D_{KL}(\pi_{tar}^{t+1}(\cdot|s)||\widetilde{\pi}_g^t(\cdot|s)) \geq \frac{1}{2\ln 2}||\pi_{tar}^{t+1}(\cdot|s) - \widetilde{\pi}_g^t(\cdot|s)||_1^2$, where $||\cdot||_1$ is the L1 norm. So

we have that for all $s \in \mathcal{S}$, $||\pi_{tar}^{t+1}(\cdot|s) - \widetilde{\pi_g^t}(\cdot|s)||_1 \leq \sqrt{2\ln 2\delta}$. According to the Performance Difference Lemma (Kakade & Langford, 2002), we have that for all $s \in \mathcal{S}$:

$$
\begin{aligned}
& V_{\widetilde{\pi_g^t}}(s) - V_{\pi_{tar}^{t+1}}(s) \\
& = \frac{1}{1-\gamma} \mathbb{E}_{s' \sim \mu_s^{\widetilde{\pi_g^t}}(s')} \left[ EA_{\pi_{tar}^{t+1}}(s, \widetilde{\pi_g^t}) \right] \\
& \leq \frac{1}{1-\gamma} \max_{s' \in \mathcal{S}} \left[ \mathbb{E}_{a \sim \widetilde{\pi_g^t}(\cdot|s')}[Q_{\pi_{tar}^{t+1}}(s', a)] - \mathbb{E}_{a \sim \pi_{tar}^{t+1}(\cdot|s')}[Q_{\pi_{tar}^{t+1}}(s', a)] \right] \\
& = \frac{1}{1-\gamma} \max_{s' \in \mathcal{S}} \int \left( \widetilde{\pi_g^t}(\cdot|s) - \pi_{tar}^{t+1}(a|s) \right) Q_{\pi_{tar}^{t+1}}(s', a) da \\
& \leq \frac{1}{1-\gamma} \max_{s' \in \mathcal{S}} \int \left| \widetilde{\pi_g^t}(a|s) - \pi_{tar}^{t+1}(a|s) \right| \cdot \left| Q_{\pi_{tar}^{t+1}}(s', a) \right| da \\
& \leq \frac{1}{1-\gamma} \max_{s' \in \mathcal{S}} \int \left| \widetilde{\pi_g^t}(a|s) - \pi_{tar}^{t+1}(a|s) \right| \cdot \frac{\widetilde{R}_{max}}{1-\gamma} da \\
& = \frac{\widetilde{R}_{max}}{(1-\gamma)^2} \max_{s' \in \mathcal{S}} ||\widetilde{\pi_g^t}(\cdot|s) - \pi_{tar}^{t+1}(\cdot|s)||_1 \\
& \leq \frac{\sqrt{2\ln 2\delta} \widetilde{R}_{max}}{(1-\gamma)^2}.
\end{aligned}
$$
(21)

Note that $|Q_\pi(s,a)| = |\mathbb{E}_\pi[\sum_{i=0}^\infty \gamma^t r(s_{t+i}, a_{t+i})|s_t = s, a_t = a]| \leq \mathbb{E}_\pi[\sum_{t=0}^\infty \gamma^t \widetilde{R}_{max}] = \frac{\widetilde{R}_{max}}{1-\gamma}$. Eventually, we have $V_{\pi_{tar}^{t+1}}(s) \geq V_{\widetilde{\pi_g^t}}(s) - \frac{\sqrt{2\ln 2\delta} \widetilde{R}_{max}}{(1-\gamma)^2} \geq V_{\pi_{tar}^t}(s) - \frac{\sqrt{2\ln 2\delta} \widetilde{R}_{max}}{(1-\gamma)^2} - \frac{2\epsilon}{1-\gamma}$.

## C   Discussion on the Influence of Over-Estimation

As CUP takes an argmax over expected Q values, it may suffer from the value over-estimation issue in DRL (Ostrovski et al., 2021). Although CUP may over-estimate values on rarely selected actions, this over-estimation serves as a kind of exploration mechanism, encouraging the agent to explore actions suggested by the source policies and potentially improving the learning target policy. If the source policies give unsuitable actions, then after exploration this over-estimation is resolved and these unsuitable actions will not be selected again. Results in Figure 8(b) suggest that even if all source policies are random and do not give useful actions, CUP still performs similarly to the original SAC, and is almost unaffected by the over-estimation issue, as over-estimation is addressed after exploring these actions.

## D   Experimental Settings

### D.1   Additional Implementation Details

To improve CUP's computation efficiency, we store the source polices' output $\{\pi_1(\cdot|s), \pi_2(\cdot|s), ..., \pi_n(\cdot|s)\}$ in the replay buffer. As we can query source policies with batches of states, and each state in the buffer only need to be queried for once, CUP is computationally efficient. Empirically, CUP only takes about 30% more wall-clock time than SAC to run the same number of environment steps. All experiments are run on GeForce GTX 2080 GPUs. The policy regularization is added after 0.5M environment environment steps to achieve more stable learning.

SAC utilizes two Q functions to mitigate the overestimation error. When CUP forms the guidance policy, we use the max value of the two target Q functions to estimate the expected advantage, which contributes to bolder exploration. Using target networks contributes to more stable training.

Equation 5 requires estimating expectations over Q values. In practice, to be efficient, we estimate the expectation by sampling a few actions (e.g., 3 actions) from each action probability distribution proposed by the source policies, and find it sufficient to achieve stable performance.

Table 1: Detailed hyper-parameter settings for CUP.

| Hyper-Parameter | Hyper-Parameter Values |
|---|---|
| batch size | 1280 |
| non-linearity | ReLU |
| actor/critic network structure | fully connected networks, three fully connected layers with 400 units |
| policy initialization | standard Gaussian |
| exploration parameters | run a uniform exploration policy 50k steps |
| learning rates for all networks | 3e-4 |
| # of samples / # of train steps per iteration | 10 env steps / 1 training step |
| optimizer | adam |
| Episode length (horizon) | 500 |
| beta for all optimizers | (0.9, 0.999) |
| discount | 0.99 |
| reward scale | 1.0 |
| temperature | learned |
| # of environment steps before adding KL regularization | 500k |
| $beta_1$ | 30 |
| $\beta_2$ | 3e-3 |

As for HAAR, we fix the source policies, and train a high-level policy as well as an additional low-level policy with HAAR's auxiliary rewards.

## D.2 Hyper-Parameter Details

All hyper-parameters used in our experiments are listed in Table 1. We use the same set of hyper-parameters for all six tasks. We also use the same set of hyper-parameters for both CUP and the SAC baseline. Most hyper-parameters are adopted from Sodhani et al. (2021).

## D.3 Discussions on Hyper-Parameter Design

CUP has two additional hyper-parameters compared to SAC, $\beta_1$ and $\beta_2$. We provide some insight on choosing $\beta_1$ and $\beta_2$. Note that the maximum weight for the KL regularization is $\beta_1 * \beta_2 |\widetilde{V}_{\pi^t_{tar}}(s)|$, and the original actor loss $L_{actor}$ has roughly the same magnitude as $|\widetilde{V}_{\pi^t_{tar}}(s)|$. So $\beta_1 * \beta_2$ roughly determines the maximum regularization weight. Following previous works on regularization [1,2], (0.1, 1) is a reasonable range for $\beta_1 * \beta_2$. As a consequence, we choose (0.04, 1) as the range of $\beta_1 * \beta_2$ for our hyper-parameter ablation studies. What's more, as $\beta_2$ upper bounds the maximum confidence on the expected advantage estimation (Section 3.2), $\beta_2$ should be decreased if a large variance in performance is observed. These two insights efficiently guide the design of $\beta_1$ and $\beta_2$. As shown in Section 4.3, CUP achieves stable performance on a large range of $\beta_1$s and $\beta_2$s.

# E  Additional Experiment Results

## E.1  Success Rate Evaluation

Fig. 9 and Fig.10 shows performance evaluated by success rates. The performance is consistent with performance evaluated by cumulative return.

## E.2  Guidance Policy Analysis for All Tasks

This subsection provides analyses of guidance policies on all six tasks, as shown in Fig. 11 and Fig. 12. Results demonstrate that the Pick-Place source policy is the most useful in Shelf-Place, Hammer, and Pick-Place-Wall, while the Push source policy is the most useful in Push-Back and Push-Wall. In Peg-Insert-Side, both source policies are of similar usefulness.

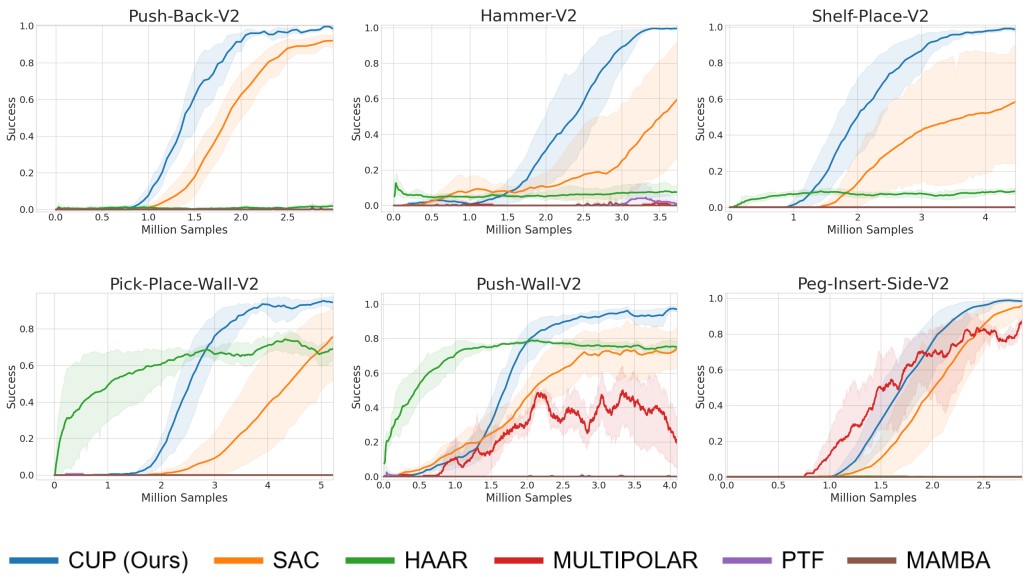

Figure 9: Algorithm performance evaluated by success rate (three source policies).

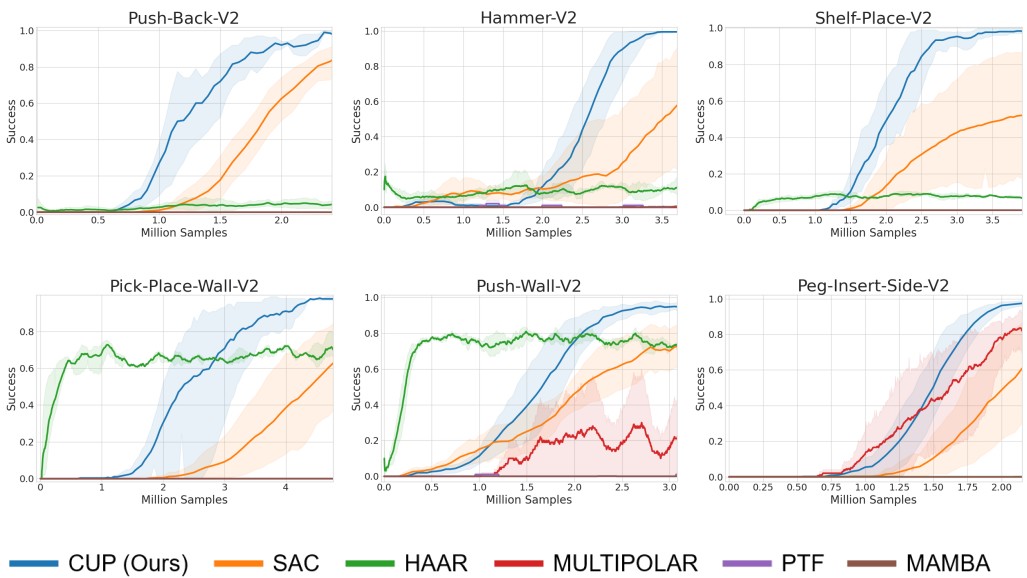

Figure 10: Algorithm performance evaluated by success rate (six source policies).

### E.3 CUP's Ability to Use Additional Source Policies

In Fig. 6 the improvement brought by additional source policies is mild, because the original three source policies have already provided sufficient support for policy reuse, as demonstrated in Fig. 13. To evaluate CUP's ability to utilize additional source policies, we design another two sets of source policies. Set 1 consists of three source policies that solve Reach, Peg-Insert-Side, and Hammer, while Set 2 adds source policies trained on Push-Back, Pick-Place-Wall, and Shelf-Place to Set 1. As Set 1 is less related to our target task Push-Wall, CUP must utilize the additional source policies in Set 2 to improve its performance. As demonstrated in Fig. 14, CUP can efficiently take advantage of the additional source policies to achieve efficient learning.

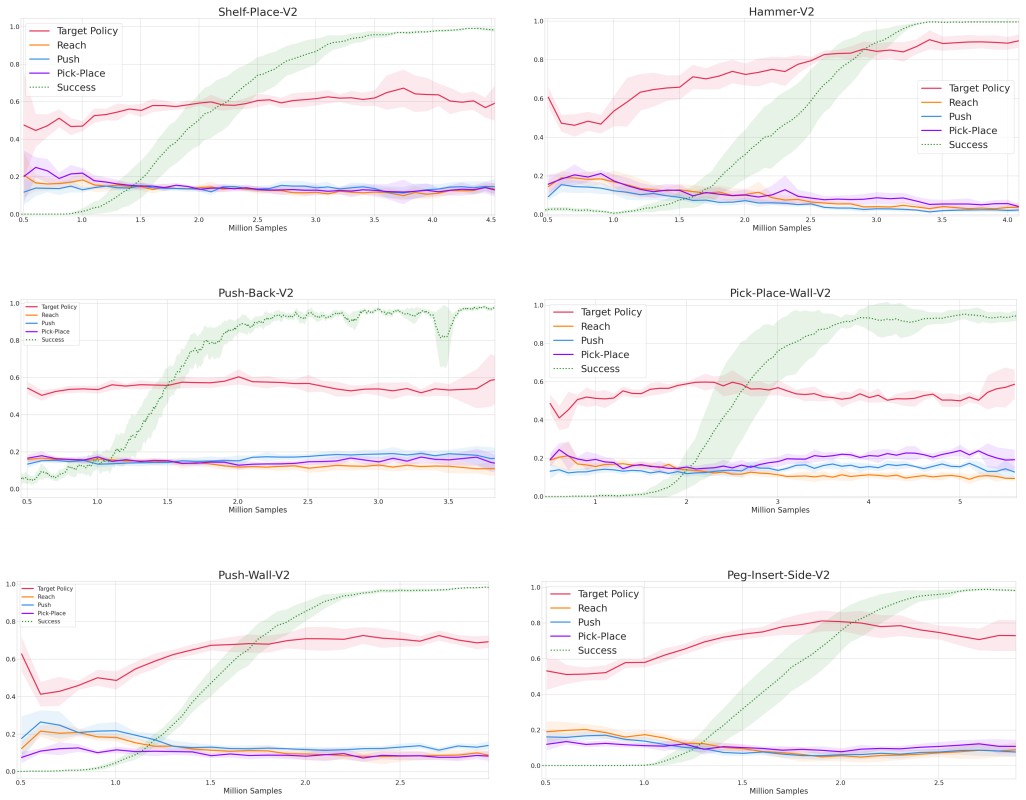

Figure 11: Percentages of source policies being selected by CUP during training on all six tasks. The green dashed line represents the target policy's success rate on the task.

## E.4 CUP's Source Policy Selection on the Source Task

To further investigate CUP's formation of the guidance policy, we train CUP on one of the source tasks, Push. As shown in Fig. 15, the corresponding source policy Push is selected frequently. After the target policy converges, CUP selects the target policy and the Push policy for roughly the same frequency, as they can both solve the task.

## E.5 Experiments on Additional Tasks

We evaluate CUP on Bin-Picking and Stick-Pull, two tasks less related to the source policies. As demonstrated in Fig. 16, in this harder setting, CUP's performance improvement over SAC is smaller. To investigate this, we provide an analysis on CUP's source policy selection. As shown in Fig. 17, the smaller improvement is because that source policies are less related to the target tasks, as they are selected less frequently.

## E.6 Analyzing Non-Stationarity in HRL Methods

To further analyze the advantages of CUP and demonstrate the non-stationarity problem of HRL methods, we illustrate the percentages of each low-level policy being selected by HAAR's high-level policy. HAAR's low-level policy set consists of the three source policies and an additional trainable low-level policy, which is expected to be selected at states where no source policies give useful actions. As demonstrated in Fig. 18(a) and Fig. 18(b), HAAR's low-level policy selection suffers from a large variance over different random seeds, and oscillates over time. This is because that as the low-level policy keeps changing, the high-level transition becomes non-stationary and leads to unstable learning. In comparison, as shown in Fig. 18(c) and Fig. 18(d), CUP's source policy selection is much more stable and achieves superior performance, as it selects source policies

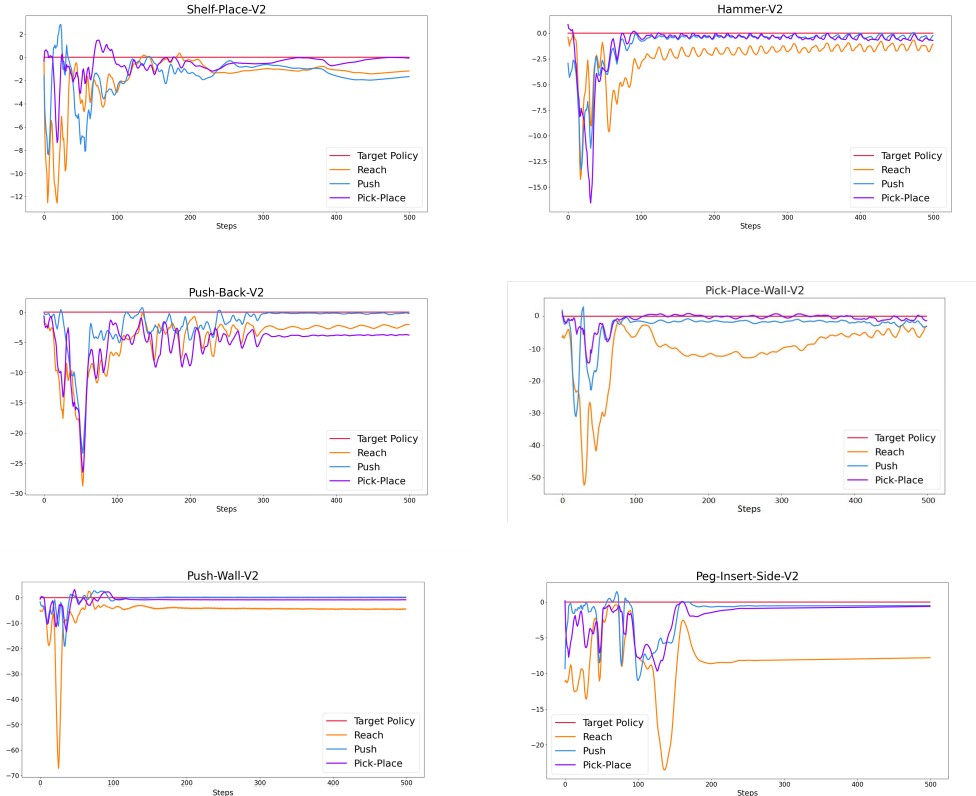

Figure 12: Expected advantages of source policies at convergence on all six tasks. The horizontal axis represents the environment steps of an episode.

according to expected advantages instead of high-level policies, and avoids the non-stationarity problem.

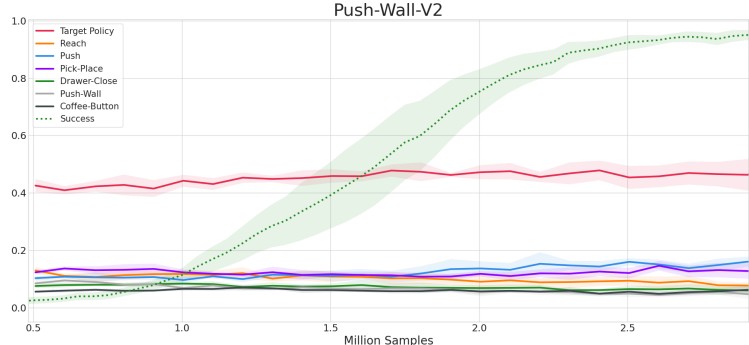

Figure 13: Percentages of source policies being selected by CUP during training on Push-Wall with the original set of six source policies. The additional source policies are seldom selected, which suggests that the first three source policies already provide sufficient support for policy reuse.

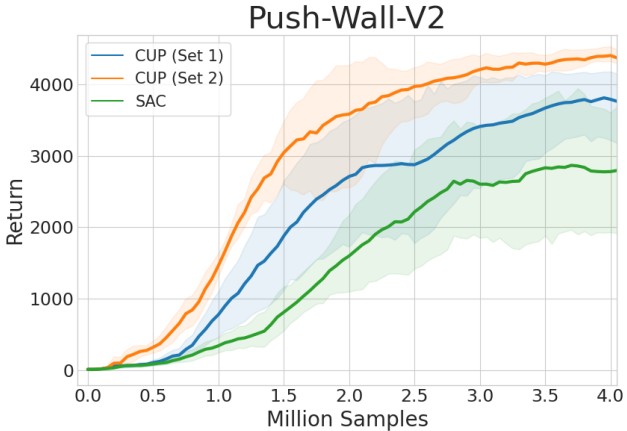

Figure 14: CUP's performance on another two sets of source policies. CUP can efficiently utilize the additional source policies contained in Set 2.

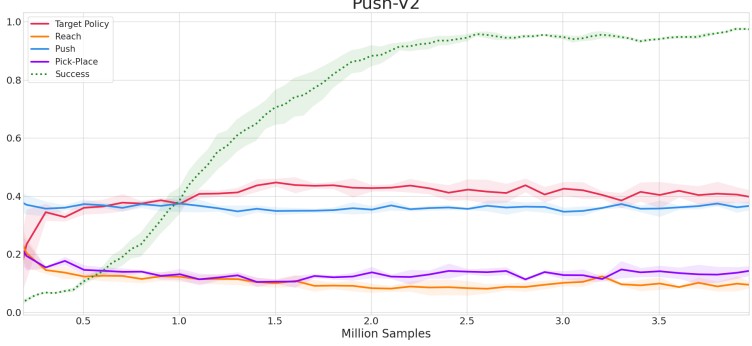

Figure 15: Percentages of source policies being selected by CUP during training on Push. The green dashed line represents the target policy's success rate on the task. The Push source policy is selected far more frequently than the other two source policies, and is selected for roughly the same frequency as the target policy at convergence.

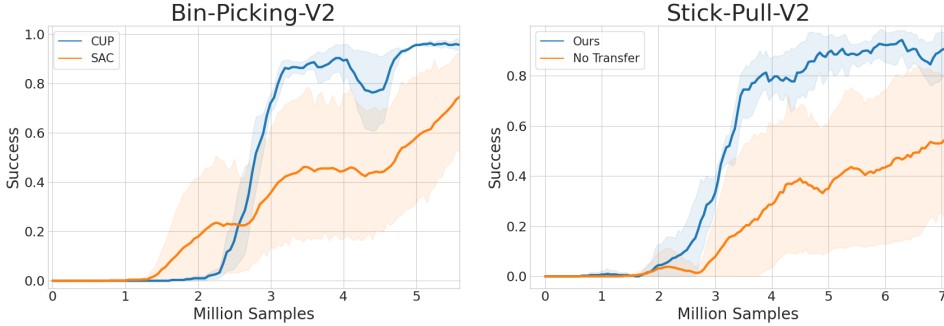

Figure 16: Performance of CUP and SAC on two harder Meta-World tasks that require more environment steps to converge.

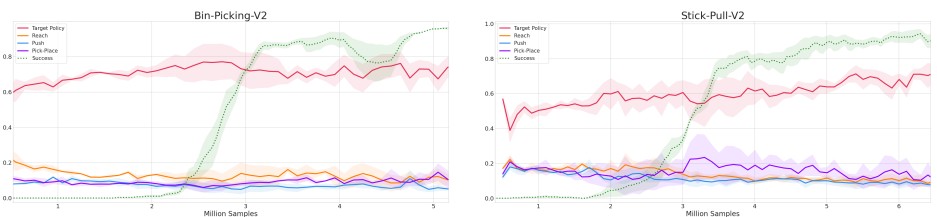

Figure 17: Percentages of source policies being selected by CUP during training. In these two tasks, the source policies are chosen less frequently, which implies that source policies are less related to these tasks.

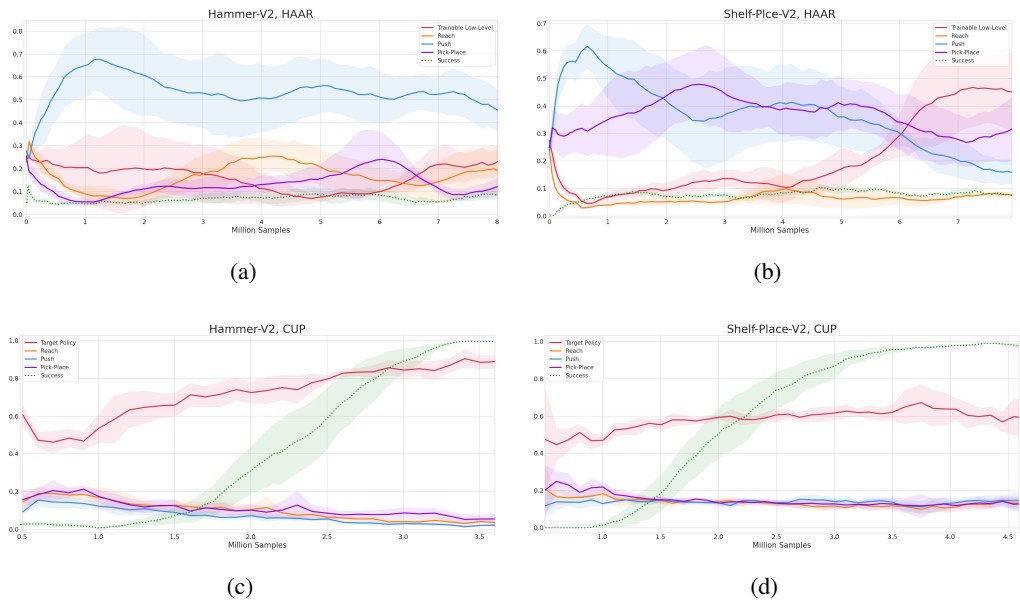

Figure 18: Comparison between HAAR and CUP's source policy selection on two representative tasks. Results are averaged over six random seeds. (a) and (b) demonstrates the percentages of each low-level policy being selected by HAAR's high-level policy. "Trainable Low-Level" is HAAR's additional trainable low-level policy, as mentioned in Appendix E.6. (c) and (d) demonstrates the the percentages of each source policy being selected by CUP. While HAAR suffers from the non-stationarity problem and has a large variance in source policy selection, CUP is much more stable and achieves superior performance, as CUP avoids the non-stationarity problem by avoiding training high-level policies.