# OpenReview forum: "CUP: Critic-Guided Policy Reuse"
_NeurIPS.cc/2022/Conference — NeurIPS 2022 Accept_

### Official Review · Reviewer_HzuR · 2022-07-04

**Rating:** 6
**Confidence:** 5
**Soundness:** 3 good
**Presentation:** 4 excellent
**Contribution:** 3 good

**Summary:**

This paper is meant to achieve efficient policy reuse for resolving complex tasks. Specifically, they introduce Critic-gUided Policy reuse (CUP), evaluating and choosing appropriate source policies to regularize the training on the target tasks. Experiments shows convincing improvements over kinds of baselines


**Questions:**

1. You mentioned that “MULTIPOLAR fails in more complex tasks” but the algorithm works in the last two figures. So what is the order of difficulty of these six tasks?

2. Why not provide the percentages and expected advantages figure for all tasks (in the Appendix)? Better include all figures in the Appendix for completeness.

3. It seems the proposed method reuses the source policies by distilling them into a single one. Can the author provide more discussion about the advantage (e.g., stability, intuition, efficiency) to do so compared to those HRL works who learned to choose different source policies? Why distill all knowledge into a single policy a better idea?


**Limitations:**

The authors have addressed their limitations.

**Strengths And Weaknesses:**

Strength:
The topic of reusing simple source policies for resolving complex tasks is important. The paper is clearly stated, well written, and easy to follow. The proposed method is intuitive, simple and straightforward. The experiment comparison is strong and convincing.

Weakness:
- It seems that the proposed method tends to be affected by the chosen of hyper-parameters. Although the authors show in Figure 5 that “CUP performs well on a wide range of hyper-parameters”, it is not quite a large range. From the formulation, I find it will be a little tricky to tune the two $\beta$s to reach a balanced imitation.
- List all hyper-parameters used in your experiments and provide both default and more guidance to the hyper-parameter settings will help relieve this concern.

I am willing to vote for an accept to this paper, but I would like to do so after the authors can relieve my concern about the hyper parameters and completeness (also see Questions below).

===

After the first round of rebuttal, the author addressed most of my concerns, and I am increasing my score to 5.

===

After the second round of rebuttal, the author further addressed my concerns, and I am increasing my score to 6.

---

> ### Author Response · Authors · 2022-08-02
> **Response to Reviewer HzuR (Part 1)**
>
> Thank you for the insightful comments. We provide clarification to your questions and concerns as below. We appreciate any further questions or comments.
>
> **Q1.** It seems that the proposed method tends to be affected by the chosen of hyper-parameters.?From the formulation, I find it will be a little tricky to tune the two $\beta$s to reach a balanced imitation.
>
> **A1.** To show CUP's robustness on hyper-parameters, we provide additional results in Figure 10 in Appendix A.7. We test CUP on a wider range of hyper-parameters. Results demonstrate that CUP achieves stable performance even if $\beta_1$ and $\beta_2$ are more than three times as large as their default values. On the other hand, as stated in Section 4.3.1, CUP uses the same set of hyper-parameters for all the six tasks presented in the paper, which also suggests that CUP is robust to the choice of hyper-parameters.
>
> Here are some insights on choosing $\beta_1$ and $\beta_2$. Note that the maximum weight for the KL regularization is $\beta_1$\*$\beta_2$\*|$\tilde{V}$\_{tar}^{t}(s)| , and the original actor loss $L_{actor}$ has roughly the same magnitude as |$\tilde{V}$\_{tar}^{t}(s)|. So $\beta_1$\*$\beta_2$ roughly determines the maximum regularization weight. Following previous works on regularization [1,2], (0.1, 1) is a reasonable range for $\beta_1$\*$\beta_2$. As a consequence, we choose (0.04, 1) as the range of $\beta_1*\beta_2$ for our hyper-parameter ablation studies. What's more, as $\beta_2$ upper bounds the maximum confidence on the expected advantage estimation (Section 3.2), $\beta_2$ should be decreased if a large variance in performance is observed. These two insights efficiently guide the design of $\beta_1$ and $\beta_2$.
>
> **Q2.** List all hyper-parameters used in your experiments and provide both default and more guidance to the hyper-parameter settings will help relieve this concern.
>
>
> **A2.** CUP has two additional hyper-parameters compared to SAC, $\beta_1$ and $\beta_2$. We provide their default values in Appendix A.4, which are used in all six tasks. We discuss their robustness and design choices in **A1.**. We adopt the default parameters for SAC from [3]. All hyper-parameters are listed in Table 1 in Appendix A.8.
>
>
>
>
> **Q3.** You mentioned that¡°MULTIPOLAR fails in more complex tasks¡±but the algorithm works in the last two figures. So what is the order of difficulty of these six tasks?
>
> **A3.** For the problem of policy reuse, task difficulty is generally determined by two factors: usefulness of source policies on the target task, and difficulty of learning the target policy on states where source policies are not useful. MULTIPOLAR works on Push-Wall-V2 and Peg-Insert-Side-V2, because the Push source policy is useful on Push-Wall-V2 (implied by HAAR's good jump-start performance), and learning residuals (discussed in Section 5) on Peg-Insert-Side-V2 is easier (implied by SAC's fast learning). In Pick-Place-Wall-V2, the Pick-Place source policy is useful, but the residual is difficult to learn, so MULTIPOLAR does not work. For the remaining three tasks, the source policies are less useful, so they are more difficult.
>
> **Q4.** Why not provide the percentages and expected advantages figure for all tasks (in the Appendix)? Better include all figures in the Appendix for completeness.
>
>
>
> **A4.** We have added all the percentages and expected advantages figures for all tasks in Appendix A.9. These results accord to our original analysis in Section 4.2, and reflect the usefulness of source policies on target tasks.
>
>
> **Q5.** It seems the proposed method reuses the source policies by distilling them into a single one. Can the author provide more discussion about the advantage (e.g., stability, intuition, efficiency) to do so compared to those HRL works who learned to choose different source policies? Why distill all knowledge into a single policy a better idea?
>
> **A5.** CUP does not explicitly distill source policies into a single policy network. Instead, in each iteration, CUP forms a guidance policy that is a dynamic aggregation of source policies and the current target policy by querying the current critic. The guidance policy is theoretically guaranteed to be a monotonic improvement over the current target policy.
>
> As discussed in the second paragraph in Section 1, HRL methods suffer from non-stationarity issues, as they require jointly training of high-level and low-level policies [4,5]. One intuition for CUP is to find another way to decide which source policy to reuse rather than training high-level policies. Using expected advantages to reuse source policies is conceptually simple, easy to implement, has theoretical guarantees, and avoids the non-stationarity problem.

---

> > ### Author Response · Authors · 2022-08-02
> > **Response to Reviewer HzuR (Part 2)**
> >
> > ### Reference
> >
> > [1] Van Laarhoven, T. (2017). L2 regularization versus batch and weight normalization. arXiv preprint arXiv:1706.05350.
> >
> > [2] Wu, Y., Tucker, G., & Nachum, O. (2019). Behavior regularized offline reinforcement learning. arXiv preprint arXiv:1911.11361.
> >
> > [3]Sodhani, S., Zhang, A., & Pineau, J. (2021, July). Multi-task reinforcement learning with context-based representations. In International Conference on Machine Learning (pp. 9767-9779). PMLR.
> >
> > [4] Pateria, S., Subagdja, B., Tan, A. H., & Quek, C. (2021). Hierarchical reinforcement learning: A comprehensive survey. ACM Computing Surveys (CSUR), 54(5), 1-35.
> >
> > [5] Li, S., Wang, R., Tang, M., & Zhang, C. (2019). Hierarchical reinforcement learning with advantage-based auxiliary rewards. Advances in Neural Information Processing Systems, 32.

---

> > ### Comment · Reviewer_HzuR · 2022-08-05
> > **Thanks for addressing my concerns.**
> >
> > I've read all responses and have seen the revised paper, it is now more clear and more complete paper to be accepted and the author has addressed most of my concerns. I am now increasing my score to 5, but I still have questions to discuss with the authors for potential improvements.
> >
> >
> > - `CUP does not explicitly distill source policies into a single policy network`, I understand this is not an explicit distillation, but it seems like a curriculum way of implicit distillation? Furthermore, you said `HRL methods suffer from non-stationarity issues`, this seems not well-justified. Did you mean that CUP `avoids the non-stationarity problem` by `monotonic improvement`?
> >
> > - `another way to decide which source policy to reuse rather than training high-level policies` is good, but can the authors put more evidences on the advantage of CUP compared with HRL methods? Along with the above question, if additional discussions/experiments can be conducted in the Appendix it will be much more convincing and fully discussed.
> >
> > - Can you put the "insights on choosing $\beta1$ and $\beta2$" along with the description of task difficulty into your paper? It will helps to understand the proposed algorithm.
> >
> > - A slight advice for formating your appendix: you can format your appendix by concluding similar subsections into a separate appendix section, for example, `Appendix A Proofs; Appendix B Experimental Settings; Appendix C Additional Results` (for example like the PPO paper does). The current appendix is complete, though, it seems a little bit hard reading. Just kindly reminds.
> >
> > I am willing further increase my score if the author can further improve their paper into a better version.

---

> > > ### Author Response · Authors · 2022-08-06
> > > **Further Response**
> > >
> > > Thank you so much for the prompt feedback and thoughtful advice. We are pleased that our previous response has addressed most of your concerns. Further clarification and discussion about the remaining questions are provided as follows.
> > >
> > > **Q1.** CUP seems like a curriculum way of implicit distillation.
> > >
> > > **A1.** Yes, your understanding is sensible. CUP is a dynamic implicit distillation, as the aggregation is based on the current critic as well as the current target policy, which are gradually improving during learning.
> > >
> > > **Q2.** (1) You said `HRL methods suffer from non-stationarity issues`, this seems not well-justified. (2) Did you mean that CUP `avoids the non-stationarity problem` by `monotonic improvement`?
> > >
> > > **A2.** (1) The non-stationarity issue is a common challenge in the field of HRL [1,2,3,4,5]. Many prior HRL works have proposed methods to alleviate this problem, such as designing intrinsic rewards [2], using off-policy corrections [3], adding regularizations to subgoal representations [4], and using hindsight transitions [5]. We provide further empirical evidence on the non-stationarity issue in **A3.** below.
> > >
> > > (2) CUP `avoids the non-stationarity problem` by avoiding training high-level policies, and instead uses expected advantages to choose source policies. The `monotonic improvement` is a property of CUP, and is not a direct reason why CUP avoids the non-stationarity problem.
> > >
> > > **Q3.** Can the authors put more evidence on the advantage of CUP compared with HRL methods?
> > >
> > > **A3.**  As demonstrated in Figure 2, two SOTA HRL algorithms, HAAR and PTF, both significantly underperforms CUP in Meta-World benchmark tasks. To further analyze the advantages of CUP and demonstrate the non-stationarity problem of HRL methods, we illustrate the percentages of each low-level policy being selected by HAAR's high-level policy in two representative tasks, as shown in Figure 18(a) and Figure 18(b) in Appendix A.15, respectively. Results show that HAAR's low-level policy selection suffers from a large variance over different random seeds, and oscillates over time. This is because that as the low-level policy keeps changing, the high-level transition becomes non-stationary and leads to unstable learning. In comparison, as shown in Fig. 18(c) and Figure 18(d), CUP's source policy selection is much more stable, as it selects source policies according to expected advantages instead of high-level policies, and avoids the non-stationarity problem.
> > >
> > > **Q4.** Can you put the "insights on choosing $\beta_1$ and $\beta_2$" along with the description of task difficulty into your paper?
> > >
> > > **A4.** Thank you very much for your advice on improving the clarity of the paper. To facilitate current discussions during the rebuttal phase, we temporarily keep the current paper structure for consistency. We will move these discussions about choosing $\beta_1$ and $\beta_2$ as well as task difficulty descriptions to the main paper in the final version of our paper.
> > >
> > > **Q5.** You can format your appendix by concluding similar subsections into a separate appendix section.
> > >
> > > **A5.** Thank you very much for your advice on improving the presentation of the paper. To facilitate current discussions during the rebuttal phase, we temporarily keep the current paper structure for consistency. We will re-format our appendices in the final version of our paper.
> > >
> > > We hope our responses address your concerns. We appreciate any further feedback.
> > >
> > >
> > > ### Reference
> > >
> > > [1] Hutsebaut-Buysse, M., Mets, K., & Latré, S. (2022). Hierarchical Reinforcement Learning: A Survey and Open Research Challenges. Machine Learning and Knowledge Extraction, 4(1), 172-221.
> > >
> > > [2] Li, S., Wang, R., Tang, M., & Zhang, C. (2019). Hierarchical reinforcement learning with advantage-based auxiliary rewards. Advances in Neural Information Processing Systems, 32.
> > >
> > > [3] Nachum, O., Gu, S. S., Lee, H., & Levine, S. (2018). Data-efficient hierarchical reinforcement learning. Advances in neural information processing systems, 31.
> > >
> > > [4] Li, S., Zhang, J., Wang, J., Yu, Y., & Zhang, C. (2021, September). Active Hierarchical Exploration with Stable Subgoal Representation Learning. In International Conference on Learning Representations.
> > >
> > > [5] Levy, A., Konidaris, G., Platt, R., & Saenko, K. (2018, September). Learning Multi-Level Hierarchies with Hindsight. In International Conference on Learning Representations.

---

> > > > ### Comment · Reviewer_HzuR · 2022-08-08
> > > > **Thanks for further discussions**
> > > >
> > > > I thank the author for the further discssions to address my concern. Overall, I think the method is reasonable and interesting and now it seems more complete to be accepted. Now I am rasing my score. However there is one more question, it seems not serious to consider HAAR and PTS as the SOTA method of HRL? They are works that published at 2019 and 2020, and I believe there are more HRL paper in recent years.
> > > >
> > > > Finally, I highly suggest the authors to revise include all discussions in their final version to help readers understanding the method.

---

> > > > > ### Author Response · Authors · 2022-08-08
> > > > > **Thank You for the Prompt Feedback**
> > > > >
> > > > > Thank you very much for the positive re-evaluation and prompt feedback. We provide clarifications to your further questions as below.
> > > > >
> > > > > **Q1.** It seems not serious to consider HAAR and PTF as the SOTA method of HRL. They are works that published at 2019 and 2020, and I believe there are more HRL papers in recent years.
> > > > >
> > > > > **A1.** In recent years, considerable progress has been made in the field of HRL [1-13]. Most of the works do not focus on solving the problem of policy reuse. Instead, they solve problems such as exploration [1,2,3], subgoal representation learning [1,4], unsupervised skill discovery [5,6,7], decomposing complex tasks via subgoals [8,9], learning hierarchical policies from offline datasets [10,11], and learning options [12,13]. We have made our best on literature survey, but there is still a chance that we may miss related works. We appreciate it if you can provide more recent works on solving the problem of policy reuse with HRL.
> > > > >
> > > > > **Q2.** I highly suggest the authors to revise include all discussions in their final version to help readers understanding the method.
> > > > >
> > > > > **A2.** Thank you for your kind advice. We are thankful that reviewers have raised many valuable questions which help improve the paper, and we will include these discussions in the final version of the paper.
> > > > >
> > > > >
> > > > >
> > > > > ### Reference
> > > > >
> > > > >
> > > > >
> > > > >
> > > > > [1] Li, S., Zhang, J., Wang, J., Yu, Y., & Zhang, C. (2021, September). Active Hierarchical Exploration with Stable Subgoal Representation Learning. In International Conference on Learning Representations.
> > > > >
> > > > > [2] Gehring, J., Synnaeve, G., Krause, A., & Usunier, N. (2021). Hierarchical skills for efficient exploration. Advances in Neural Information Processing Systems, 34, 11553-11564.
> > > > >
> > > > > [3] Bagaria, A., Senthil, J. K., & Konidaris, G. (2021, July). Skill discovery for exploration and planning using deep skill graphs. In International Conference on Machine Learning (pp. 521-531). PMLR.
> > > > >
> > > > > [4] Li, S., Zheng, L., Wang, J., & Zhang, C. (2020, September). Learning subgoal representations with slow dynamics. In International Conference on Learning Representations.
> > > > >
> > > > > [5] Kim, J., Park, S., & Kim, G. (2021, July). Unsupervised Skill Discovery with Bottleneck Option Learning. In International Conference on Machine Learning (pp. 5572-5582). PMLR.
> > > > >
> > > > > [6] Zhang, J., Yu, H., & Xu, W. (2020, September). Hierarchical Reinforcement Learning by Discovering Intrinsic Options. In International Conference on Learning Representations.
> > > > >
> > > > > [7] Fang, K., Zhu, Y., Savarese, S., & Fei-Fei, L. (2021). Discovering Generalizable Skills via Automated Generation of Diverse Tasks. arXiv preprint arXiv:2106.13935.
> > > > >
> > > > > [8] Kim, J., Seo, Y., & Shin, J. (2021). Landmark-guided subgoal generation in hierarchical reinforcement learning. Advances in Neural Information Processing Systems, 34, 28336-28349.
> > > > >
> > > > > [9] Gürtler, N., Büchler, D., & Martius, G. (2021). Hierarchical reinforcement learning with timed subgoals. Advances in Neural Information Processing Systems, 34, 21732-21743.
> > > > >
> > > > > [10] Rao, D., Sadeghi, F., Hasenclever, L., Wulfmeier, M., Zambelli, M., Vezzani, G., ... & Heess, N. (2021, September). Learning transferable motor skills with hierarchical latent mixture policies. In International Conference on Learning Representations.
> > > > >
> > > > > [11] Ajay, A., Kumar, A., Agrawal, P., Levine, S., & Nachum, O. (2020, September). OPAL: Offline Primitive Discovery for Accelerating Offline Reinforcement Learning. In International Conference on Learning Representations.
> > > > >
> > > > > [12] Araki, B., Li, X., Vodrahalli, K., DeCastro, J., Fry, M., & Rus, D. (2021, July). The logical options framework. In International Conference on Machine Learning (pp. 307-317). PMLR.
> > > > >
> > > > > [13] Veeriah, V., Zahavy, T., Hessel, M., Xu, Z., Oh, J., Kemaev, I., ... & Singh, S. (2021). Discovery of options via meta-learned subgoals. Advances in Neural Information Processing Systems, 34, 29861-29873.

---

### Official Review · Reviewer_9Jza · 2022-07-09

**Rating:** 7
**Confidence:** 4
**Soundness:** 3 good
**Presentation:** 4 excellent
**Contribution:** 4 excellent

**Summary:**

CUP is an algorithm for re-using previously learned policies to guide the training of a new policy on a different-but-related task. CUP does this by selecting a single guide policy from among the library of pretrained policies at each step, which the student policy is then trained to behave similar to using a KL divergence regularization term.

**Questions:**

The conclusion section of the paper is pretty limited. I appreciate how space is limited, but some discussion of broader limitations and possible avenues for future improvement would be nice if space can be found.

In Figure 3, I'm surprised how little CUP seems to use any guide policy throughout training. I'm not sure offhand how to tap into it, but this seems like it might be a sign of leaving performance on the table? The least trained target policy is only getting updated with the KL term about half the time on a task where the push guide policy should be highly informative.

Related to that, I wonder what the percentages would be if training on one of the source tasks? For example, would the push policy get used more if training on the push task? It could provide a useful indicator for whether there's more to be gained from the source policies.

**Limitations:**

I included discussion and suggestions for limitations in the previous sections, and while I'd like to see more "hard" test cases the existing experiments do provide some idea of the limitations of CUP. The potential for negative social impact from this work is limited but is addressed in the appendix.

**Strengths And Weaknesses:**

Overall, I liked this paper. The algorithm is novel, clearly presented, and conceptually simple (a plus). The experiments provide reasonable evidence that CUP improves performance compared to both baseline and alternative teacher-student transfer algorithms.

I do have a few concerns, though I don't think any of this invalidates the results presented:

-If I understand correctly, taking an argmax among policies using a partially-trained value function seems prone to bias/error magnification. Given relatively poor Q estimates of equal magnitude for each policy's sampled actions (as can happen early in training), the guide policy selected will tend to be the one that samples actions which Q is most (unrealistically) optimistic about. Further, the estimated advantage KL term weighting makes these updates larger.
I think the authors appreciate this, hence their value function upper bound on the weighting term and not using the KL term for the first 0.5M environment steps (as per A.4), and the result is an algorithm that works in practice (as shown by the experiments). That said, it does make me wonder how well performance will hold up as the difference between source and target tasks increases (where many actions sampled by source policies will be bad for any given state). The random-source-policy ablation sort-of tests this, but still assumes a subset of source policies are relatively high-performing. Basically, can CUP be used to gain a training benefit from weak teacher policies?

-Relatedly, the bound in theorem 2 is dependent on the difference between source and target policies( as well as reward magnitude), and could be a very large bound given adversarial values. I'm willing to accept that this isn't an issue in practice (at least for Metaworld), but I'm curious to see how those factors impact empirical performance.

-Connected to the above two points, while it may be something of a stereotype for reviewers to ask for more experiments, additional experiments on other Metaworld tasks (either the full suite of 50 or select "hard" tasks that are less similar to the source tasks) would improve the paper in a worthwhile way. Ideally I'd like to have some qualitative evidence for how different aspects of source versus target tasks affect performance for CUP. In the current results it looks like CUP improves over SAC less on Hammer and Peg-Insert-Side, the two "more novel" tasks, but without more tasks or deeper analysis it's hard to say anything conclusive.

---

> ### Author Response · Authors · 2022-08-02
> **Response to Reviewer 9Jza (Part 1)**
>
> Thank you for the insightful comments. We provide clarification to your questions and concerns as below. We appreciate any further questions or comments.
>
> **Q1.** (1) Taking an argmax among policies using a partially-trained value function seems prone to bias/error magnification. (2) It does make me wonder how well performance will hold up as the difference between source and target tasks increases. Basically, can CUP be used to gain a training benefit from weak teacher policies?
>
>
>
> **A1.** (1) The reviewer raises a good point. Although CUP may over-estimate values on rarely selected actions, this over-estimation serves as a kind of exploration mechanism, encouraging the agent to explore actions suggested by the source policies and potentially improving the learning target policy. If the source policies give unsuitable actions, then after exploration this over-estimation is resolved and these unsuitable actions will not be selected again.
>
> (2) Figure 15 in Appendix A.11 demonstrates that even if all source policies are random and do not give useful actions, CUP still performs similarly to the original SAC and is almost unaffected by the over-estimation issue, as over-estimation is addressed after exploring these actions. We also add a Reach source policy to the three random source policies and test CUP on Push-Wall-V2, a task in which the Reach source policy is not high-performing. Figure 15 also demonstrates that even when there is only one less-useful source policy accompanied with three random source policies disrupting policy reuse, CUP is still able to improve learning efficiency by reusing the meaningful source policy.
>
> **Q2.**  Relatedly, the bound in theorem 2 is dependent on the difference between source and target policies (as well as reward magnitude), and could be a very large bound given adversarial values. I'm willing to accept that this isn't an issue in practice (at least for Metaworld), but I'm curious to see how those factors impact empirical performance.
>
>
> **A2.** The bound is dependent on the difference between the current target policy and the guidance policy, and it generally will not be too large because: (1) we minimize the KL divergence between the target policy and the guidance policy during training (Eq. 11), and (2) the guidance policy is an aggregation of source policies and the current target policy (Eq. 5). The reward magnitude is closely related to the value's magnitude, so the gap would not be too large.
>
>
> **Q3.** In Figure 3, I'm surprised how little CUP seems to use any guide policy throughout training.
>
> **A3.** Figure 3 illustrates the percentages of the guidance policy selecting each source policy and the current target policy. Although each single source policy does not seem to be selected very often, they sum up to be selected for about 40\% of the time. As the target policy is continuously improving and becomes more competitive, the guidance policy will gradually decrease its usage of source policies.
>
>
> **Q4.**  Related to that, I wonder what the percentages would be if training on one of the source tasks? For example, would the push policy get used more if training on the push task? It could provide a useful indicator for whether there's more to be gained from the source policies.
>
>
> **A4.** Thank you for your suggestion. We conducted an experimental as suggested by the reviewer. As demonstrated in Fig. 16, while training on Push-V2, the corresponding source policy Push is selected frequently. After the target policy converges, CUP selects the target policy and the Push policy for roughly the same frequency, as they can both solve the task.
>
>
> **Q5.**  Ideally I'd like to have some qualitative evidence for how different aspects of source versus target tasks affect performance for CUP. In the current results it looks like CUP improves over SAC less on Hammer and Peg-Insert-Side, the two "more novel" tasks, but without more tasks or deeper analysis it's hard to say anything conclusive.
>
>
>
> **A5.** The usefulness of source policies on the target task can be evaluated by the frequency of the source policies being selected by the guidance policy during training. As demonstrated in Figure 11, Hammer-V2 and Peg-Insert-Side-V2 are "more novel" tasks, as the target policies are being selected for about 80\% of the time at convergence (while for other tasks the number is about 60\%), which indicates that source policies are less useful on these two tasks. This result implies that the performance improvement bought by policy reuse is closely related to the usefulness of source policies on the target task. We also test CUP on another two "more novel" tasks, as discussed in **A6** below.

---

> > ### Author Response · Authors · 2022-08-02
> > **Response to Reviewer 9Jza (Part 2)**
> >
> > Q6. Additional experiments on other Metaworld tasks (either the full suite of 50 or select "hard" tasks that are less similar to the source tasks) would improve the paper in a worthwhile way.
> >
> > A6. We provide results on two "harder" tasks that source policies are less useful on them (supported by results in Figure 18, which illustrates the frequency of the source policies being selected by the guidance policies on these two tasks). Figure 17 in Appendix A.13 demonstrates that the effect of policy reuse decreases as source policies become less useful on target tasks. As for the full suite of Meta-World tasks, many of the tasks are so easy that learning without policy reuse is already very efficient.
> >
> > Q7. The conclusion section of the paper is pretty limited. I appreciate how space is limited, but some discussion of broader limitations and possible avenues for future improvement would be nice if space can be found.
> >
> > A7. As suggested by Reviewer nMQw and biXP, one limitation of CUP is the assumption of the source policies and the target policy sharing the same state and action spaces. We have added discussions on CUP's limitations as well as possible future directions to Appendix A.14. We will move this discussion to the main paper in the final version of our paper.

---

> ### Author Response · Authors · 2022-08-08
> **Any Further Questions or Concerns are Welcome**
>
> Dear Reviewer 9Jza,
>
> Since the author-reviewer discussion period is approaching the deadline, we would appreciate it if you could check our response to your review comments soon. This way, if you have further questions and comments, we can still reply before the author-reviewer discussion period ends. Thank you very much for your time and efforts!
>
> Best,
>
> The authors

---

> > ### Comment · Reviewer_9Jza · 2022-08-08
> > **Brief Response**
> >
> > Hi Authors,
> >
> > Thanks for your answers! I agree with your answers by and large, and it looks like you've added material to the paper to address them already.
> >
> > As a sidenote, I'm impressed at the extra experiments you've run to address some of my points! I wasn't requiring/expecting that my suggestions be implemented immediately, but I like the paper even better for having them!

---

> > > ### Author Response · Authors · 2022-08-09
> > > **Thank You for Your Encouraging Response**
> > >
> > > Thank you for the encouraging response! We are glad that our response addresses your concerns. We are grateful for your valuable questions and suggestions, which help improve the paper.

---

### Official Review · Reviewer_biXP · 2022-07-10

**Rating:** 5
**Confidence:** 4
**Soundness:** 2 fair
**Presentation:** 3 good
**Contribution:** 2 fair

**Summary:**

This paper proposes a novel policy reuse algorithm Critic-gUided Policy reuse (CUP), which avoids training any extra components and efficiently reuses source policies. CUP chooses the source policy at each state that has the largest one-step improvement over the current target policy and forms a guidance policy. The target policy can be regularized to imitate the guidance policy to perform an efficient policy search.

**Questions:**

1. In the experimental part, I don't see a noticeable improvement in the comparison of CUP’s performance with different numbers of source policies, which cannot be concluded that CUP is able to utilize the additional source policies to further improve its performance.

2. From how CUP works, it should not be affected by random policies because it chooses the largest soft expected advantage at each state $s$. As the number of random policies grows, the performance should not be affected even though the computational complexity will increase. But why CUP can only adapt to 3 random policies?

**Ethics Review Area:**

["I don’t know"]

**Limitations:**

The author should discuss the limitations of CUP. For example, CUP obeys a very strong assumption that the source policies and the target policy share the same state and action spaces, which is not common in the real world. For example, how can CUP be applied to different robot transfer settings, e.g., different types or different numbers of joints? The strong assumption limits the extension of CUP to more general scenarios.

**Strengths And Weaknesses:**

Strengths:

1. This paper not only proves that the guidance policy is guaranteed to be a monotonic improvement over the current target policy but also proves that the target policy is theoretically guaranteed to improve by imitating the guidance policy.

2. The experimental part of this paper is adequate in addition to transfer performance, including analyzing the guidance policy, the sensitivity to hyper-parameter settings and the number of source policies, and the interference with random source policies. The writing ideas of this part are also very clear.

Weaknesses:

1. Some descriptions are not clear and should be clarified. In the experimental part, I have questions about the analysis of some experimental results. For example, I don't see a noticeable improvement in the comparison of CUP’s performance with different numbers of source policies, which cannot be concluded that CUP is able to utilize the additional source policies to further improve its performance.

2. The conclusion merely repeats what was said in the introduction, lacking the limitations of CUP. For example, CUP obeys a very strong assumption that the source policies and the target policy share the same state and action spaces, which is not common in the real world. The strong assumption limits the extension of CUP to more general scenarios[1-3]. The author should add some reflections on directions for future improvements.

[1] Mutual Information Based Knowledge Transfer Under State-Action Dimension Mismatch. UAI 2020.

[2] Learning Cross-Domain Correspondence for Control with Dynamics Cycle-Consistency. ICLR 2021.

[3] Cross-domain Adaptive Transfer Reinforcement Learning Based on State-Action Correspondence. UAI 2022.

---

> ### Author Response · Authors · 2022-08-02
> **Response to Reviewer biXP**
>
> Thank you for the thoughtful comments. We provide clarification to your questions and concerns as below. We appreciate any further questions or comments.
>
> **Q1.** I don't see a noticeable improvement in the comparison of CUP's performance with different numbers of source policies, which cannot be concluded that CUP is able to utilize the additional source policies to further improve its performance.
>
> **A1.** We analyze the percentages of the six source policies being selected by CUP during learning on Push-Wall-V2. Results in Figure 13 demonstrate that the first three source policies are quite related to the target policies and provide sufficient support for policy reuse, which explains why additional source policies do not improve performance greatly. To demonstrate CUP's ability to utilize additional source policies, we design another two sets of source policies. Set 1 consists of three source policies less related to the target task Push-Wall-V2, while Set 2 adds another three more useful source policies to Set 1. As shown in Figure 14 in Appendix A.10, CUP is able to utilize the additional source policies to improve performance.
>
> **Q2.**  The conclusion merely repeats what was said in the introduction, lacking the limitations of CUP. The author should discuss the limitations of CUP.
>
>
>
> **A2.** Thank you for your insightful comments on CUP's limitations and possible future directions. We have added discussions on CUP's limitations to Appendix A.14. We will move this discussion to the main paper in the final version of our paper.
>
>
>
>
>
> **Q3.**  As the number of random policies grows, the performance should not be affected even though the computational complexity will increase. But why CUP can only adapt to 3 random policies?
>
>
>
> **A3.** With further analysis we found that, with 3 random seeds for the additional random policy experiments, the variance is large in the original results and the performance drop looks significant with the number of random policies greater than 3. In the revised paper, we have run 6 random seeds and updated Figure 5(b). Results demonstrate that adding 4 and 5 random source policies leads to a slight drop in performance. This drop is because that as the number of random policies grows, more random actions are sampled, and taking argmax over these actions' expected advantages is more likely to be affected by errors in value estimation.

---

> ### Author Response · Authors · 2022-08-08
> **Any Further Questions or Concerns are Welcome**
>
> Dear Reviewer biXP,
>
> Since the author-reviewer discussion period is approaching the deadline, we would appreciate it if you could check our response to your review comments soon. This way, if you have further questions and comments, we can still reply before the author-reviewer discussion period ends. Thank you very much for your time and efforts!
>
> Best,
>
> The authors

---

### Official Review · Reviewer_nMQw · 2022-07-10

**Rating:** 6
**Confidence:** 4
**Soundness:** 3 good
**Presentation:** 3 good
**Contribution:** 3 good

**Summary:**

This paper considers the problem of policy reusing in reinforcement learning. It assumes that there are a bunch of source policies pre-trained on related tasks. The agent is interacting with the environment to learn a target policy for the target task and hopes to make use of the available source policies. The problem is to determine when and how to use which source policy.

This paper proposes CUP to employ the critic learned on the target task to select the proper source policy in each state. To be specific, there is a set of source policies and the agent's current target policy is also considered as one possible choice in the source policy set. CUP chooses the source policy with the largest one-step improvement over the current target policy. The chosen source policy in each state together forms the guidance policy. It is theoretically proved that the value of the guidance policy can be higher than the value of the current target policy if the learned critic is accurate enough. Then the target policy is trained to imitate the guidance policy by minimizing their KL divergence. The weight of this KL divergence term in policy learning is adaptively changed during training, according to the estimated advantages of the guidance policy.

The authors conduct experiments on Meta-World and compare CUP with basic SAC, recent works HAAR, PTF, MULTIPOLAR, and MAMBA. The experimental results show the advantages of CUP. The ablative study shows that CUP is relatively robust to the choice of hyper-parameter value. Adding more source policies can be beneficial if the source policy is related to the target task.  In the set of source policies, adding up to 3 random policies does not hurt the performance of CUP, but adding 4 random policies is problematic.

**Questions:**

Could the proposed method perform robustly on settings with different sets of source policies? For example, what will happen if there are some policies really unsuitable or even harmful for the target task? Could CUP properly ignore these source policies?

Any intuitive explanation about why 4 random policies hurt the performance much in Figure 5(b)?

To get the guidance policy according to equation (7), what will happen when $\pi$ and $\pi^t_{tar}$ are very different and $\tilde{Q}_{\pi^t_{tar}}$ suffers from the over-estimation issue? For example, the target policy $\pi^t_{tar}$ may rarely select an action $a_0$ at the state $s$, so the value estimate $\tilde{Q}_{\pi^t_{tar}}(s,a_0)$ is much higher than the true value $Q_{\pi^t_{tar}}(s,a_0)$. Then the source policy selecting action $a_0$ often at the state $a$ will be chosen at this state. Yet, it may not be really beneficial for the target policy learning. This choice of source policy will hurt the sample efficiency of CUP. Do you observe this issue? Any comments about preventing it?

**Limitations:**

It seems that the authors just briefly mention one limitation in section 2. "We assume that the source policies and the target policy share the same state and action space". This assumption is widely used in prior works about policy transfer, and the authors did not propose to solve this issue. One possible choice is to learn state and action correspondence to transfer the source policy to the target state and action space, e.g., 'Learning Cross-Domain Correspondence for Control with Dynamics Cycle-Consistency'.

**Strengths And Weaknesses:**

The paper is well-organized and generally written clearly. The proposed method CUP is novel and interesting with theoretical and empirical support.

Pros:

The proposed method is technically reasonable and supported by the theoretical ground.
The evaluation is solid and analysis of CUP in ablation study help understand CUP better.

Cons:
The proposed method theoretically relies on a well-trained critic. However, the choice of source policy might be problematic if the value estimate by the critic is not accurate enough, especially when the source policy and target policy are quite different.

One critical detail is not clearly explained in the paper. How to calculate the soft estimated advantage for each source policy according to equation (4)? Getting the expectation seems not very simple given continuous action space. Then it is hard to tell whether CUP is really much more convenient than prior works using hierarchical reinforcement learning or source policy value estimation.

---

> ### Author Response · Authors · 2022-08-02
> **Response to Reviewer nMQw**
>
> Thank you for the thoughtful comments. We provide clarification to your questions and concerns as below. We appreciate any further questions or comments.
>
> **Q1.**  The proposed method theoretically relies on a well-trained critic. However, the choice of source policy might be problematic if the value estimate by the critic is not accurate enough. To get the guidance policy according to equation (7), what will happen when $\pi$ and$\pi_{tar}^{t}$ are very different and $\tilde{Q}$\_${\pi^t_{tar}}$ suffers from the over-estimation issue?
>
> **A1.**  The reviewer raises a good point. Although CUP may over-estimate values on rarely selected actions, this over-estimation serves as a kind of exploration mechanism, encouraging the agent to explore actions suggested by the source policies and potentially improving the learning target policy. If the source policies give unsuitable actions, then after exploration this over-estimation is resolved and these unsuitable actions will not be selected again. To verify this hypothesis, we show additional results in Figure 15 in Appendix A.11. These results suggest that even if all source policies are random and do not give useful actions, CUP still performs similarly to the original SAC, and is almost unaffected by the over-estimation issue, as over-estimation is addressed after exploring these actions.
>
>
> **Q2.** Could the proposed method perform robustly on settings with different sets of source policies? For example, what will happen if there are some policies really unsuitable or even harmful for the target task?
>
>
>
> **A2.** To investigate CUP's ability to ignore unsuitable source policies, we design two source policy sets: the first set consists of three random policies that are all useless for the target task, and the second set adds the Reach policy to the first set. We evaluate CUP on Push-Wall-V2. As demonstrated in Figure 15 in Appendix A.11, when none of the source policies are useful, CUP performs similarly to the original SAC, and its sample efficiency is almost unaffected by the useless source policies. If only one of the four source policies is useful, CUP can still efficiently utilize the useful source policy to improve learning performance.
>
>
>
> **Q3.** How to calculate the soft estimated advantage for each source policy according to equation (4)?
>
> **A3.** In practice, to be efficient, we estimate the expectation by sampling a few actions (e.g., 3 actions) from each action probability distribution proposed by the source policies, and find it sufficient to achieve stable performance.
>
>
>
> **Q4.**  Any intuitive explanation about why 4 random policies hurt the performance much in Figure 5(b)?
>
>
>
> **A4.**  With further analysis we found that, with 3 random seeds for the additional random policy experiments, the variance is large in the original results and the performance drop looks significant with the number of random policies greater than 3. In the revised paper, we have run 6 random seeds and updated Figure 5(b). Results demonstrate that adding 4 and 5 random source policies leads to a slight drop in performance. This drop is because that as the number of random policies grows, more random actions are sampled, and taking argmax over these actions' expected advantages is more likely to be affected by errors in value estimation.
>
> **Q5.**  It seems that the authors just briefly mention one limitation in section 2. ...... One possible choice is to learn state and action correspondence to transfer the source policy to the target state and action space, e.g., 'Learning Cross-Domain Correspondence for Control with Dynamics Cycle-Consistency'.
>
>
>
>
>
> **A5.** Thank you for your insightful comments on CUP's limitations and possible future directions. We have added discussions on CUP's limitations to Appendix A.14. We will move this discussion to the main paper in the final version of our paper.

---

> > ### Comment · Reviewer_nMQw · 2022-08-09
> > **Thank Authors for Response**
> >
> > Thank you for the detailed response! The additional clarification and experiments address my problems.

---

> ### Author Response · Authors · 2022-08-08
> **Any Further Questions or Concerns are Welcome**
>
> Dear Reviewer nMQw,
>
> Since the author-reviewer discussion period is approaching the deadline, we would appreciate it if you could check our response to your review comments soon. This way, if you have further questions and comments, we can still reply before the author-reviewer discussion period ends. Thank you very much for your time and efforts!
>
> Best,
>
> The authors

---

### Meta-Review · Area_Chair_vhXD · 2022-08-25

**Recommendation:** Accept
**Confidence:** Certain

**Metareview:**

The paper proposes a method for how to leverage a list of pretrained policies for learning a new task, by picking the guidance policy through maximal one-step policy improvement evaluated with the learned critic.

Contribution is simple, but writing, theories, and experiments/ablation studies are clean and easy to follow. There is a consensus among the reviewers for the acceptance of the paper.

Minor comments:
- adding a mechanism for automatically growing and pruning source policies could be nice extension, especially on life-long continual learning environment, where once you learned a novel-enough high-reward policy you may want to add it to the source, so when environment changes and changes back, it can reuse that learn optimal behavior. [1]
- a fun experiment to include is to ignore reward and only do imitation during policy improvement (just KL term), while still using reward critic for policy selection. If we know the source policies sufficiently cover the full optimal policy, then this could be a good debugging test.

[1] Rusu, A. A., Rabinowitz, N. C., Desjardins, G., Soyer, H., Kirkpatrick, J., Kavukcuoglu, K., ... & Hadsell, R. (2016). Progressive neural networks. arXiv preprint arXiv:1606.04671.

**Award:**

No

---

### Decision · Program_Chairs · 2022-09-14

Accept